# PolySAE: Modeling Feature Interactions in Sparse Autoencoders via Polynomial Decoding

**Panagiotis Koromilas** [1 2]   **Andreas D. Demou** [1]   **James Oldfield** [3]   **Yannis Panagakis** [2 4]   **Mihalis A. Nicolaou** [5 1]

## Abstract

Sparse autoencoders (SAEs) interpret neural network representations by decomposing activations into sparse combinations of dictionary atoms. However, SAEs assume features combine additively through linear reconstruction, an assumption that cannot capture compositional structure: linear models cannot distinguish whether "Starbucks" arises from the composition of "star" and "coffee" features or merely their co-occurrence. This forces SAEs to allocate monolithic features for compound concepts rather than decomposing them into interpretable constituents. We introduce PolySAE, which extends the SAE decoder with higher-order terms to model feature interactions while preserving the linear encoder essential for interpretability. Through low-rank tensor factorization on a shared projection subspace, PolySAE captures pairwise and triple feature interactions with small parameter overhead (3% on GPT2). Across four language models and three SAE variants, PolySAE achieves an average improvement of $\sim$8% in probing F1 while maintaining comparable reconstruction error, and produces 2–10$\times$ larger Wasserstein distances between class-conditional feature distributions. Critically, learned interaction weights exhibit negligible correlation with co-occurrence frequency ($r = 0.06$ vs. $r = 0.82$ for SAE feature covariance), suggesting that polynomial terms capture compositional structure largely independent of surface statistics. Finally, the learned interaction directions causally steer model outputs toward the corresponding compositional semantics. Code: https://github.com/pakoromilas/PolySAE

[1]The Cyprus Institute [2]University of Athens [3]University of Oxford [4]Archimedes AI/Athena Research Center [5]University of Cyprus. Correspondence to: Panagiotis Koromilas <pakoromilas@di.uoa.gr>.

*Proceedings of the 43[rd] International Conference on Machine Learning*, Seoul, South Korea. PMLR 306, 2026. Copyright 2026 by the author(s).

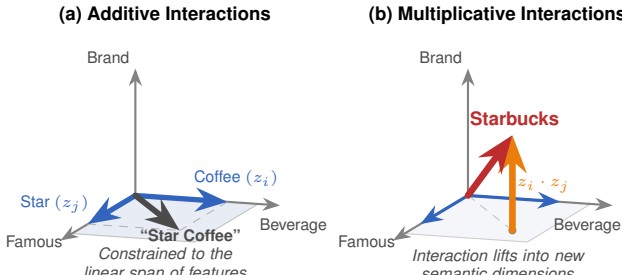

**(a) Additive Interactions**    **(b) Multiplicative Interactions**

*Figure 1.* **Semantic Dimension Expansion via Feature Interaction.** Consider two semantic directions—*Famous* and *Beverage*—and their associated learned features *Star* and *Coffee*. **(a)** Additive interactions yield co-occurrence semantics that remain in the original feature span. **(b)** Multiplicative interactions enable representations to escape this subspace via $z_i \cdot z_j$, lifting into orthogonal dimensions (*Brand*) to capture emergent concepts like *Starbucks*. "Starbucks" example from Table 4.

## 1. Introduction

As AI systems are increasingly deployed in real-world domains, ensuring their safety and reliability has become a critical challenge (Amodei et al., 2016; Hendrycks et al., 2021; Bengio et al., 2025). Developing interpretable models offers a promising path towards aligning AI with human values: understanding why a model produces a given output enables us to (i) monitor its reasoning (Lindsey et al., 2025), (ii) debug failure modes (Wong et al., 2021), and (iii) steer away from unwanted behavior (Rimsky et al., 2024). Mechanistic interpretability pursues this agenda at the level of neural network internals (Bereska & Gavves, 2024), aiming to uncover interpretable features and circuits within a model and thereby provide principled insights into its behavior.

Sparse Autoencoders (SAEs), grounded in the principles of sparse dictionary learning, have emerged as a leading tool for mechanistic interpretability. SAEs decompose neural network activations to recover human-interpretable features that models typically represent in superposition—encoded in overlapping directions due to limited representational capacity (Elhage et al., 2022). This framework has been shown to uncover safety-relevant concepts such as deception, bias, and harmful content, enabling targeted interventions that predictably steer model behavior (Templeton et al., 2024).

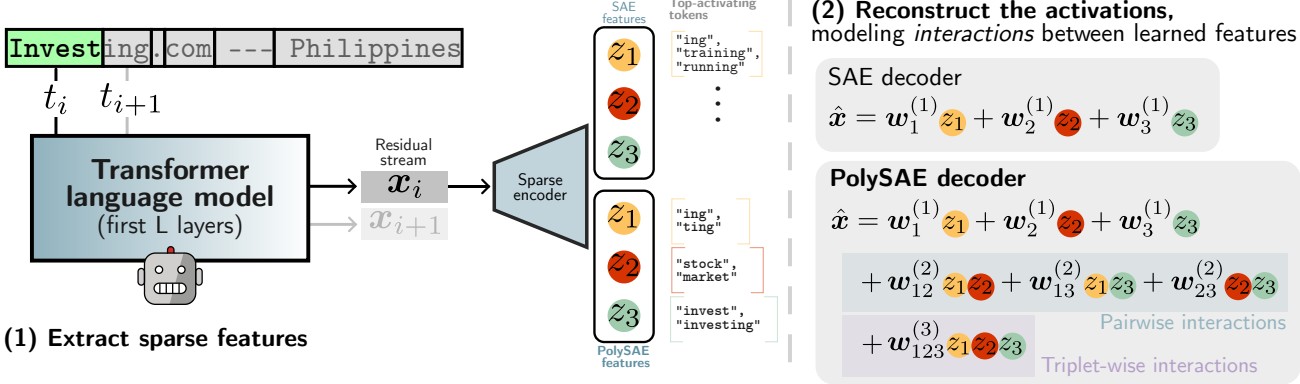

*Figure 2.* **An overview of PolySAE**: (1) sparse latent features are first extracted with a standard SAE encoder. (2) Activations in the residual stream are then reconstructed by modeling 2nd- and 3rd-order interactions in addition to the standard linear component. The example "**Invest**ing.com — Philippines stocks were higher after" comes from Table 5.

However, recent work has highlighted fundamental limitations of the SAE paradigm due to their reliance on the "strong" linear representation hypothesis (Engels et al., 2025; Csordás et al., 2024). Standard SAEs reconstruct activations as weighted sums of independent features, expressing each activation as a linear combination where features contribute additively. This linearity assumption raises a fundamental question: *what level of abstraction do learned features naturally capture?* The answer has direct implications for mechanistic interpretability. If features truly combine linearly, we would expect individual dictionary atoms to represent atomic components, such as morphemes, simple concepts, or basic semantic primitives, that combine through superposition to form complex expressions. Such atomic features would enable transparent circuit analysis and precise interventions on elemental building blocks of meaning.

Yet linguistic theory demonstrates that composition operates non-linearly across multiple levels of language structure. Morphologically, "administrators" is not simply the sum of stem and suffix; the combination produces a distinct lexical item with specific syntactic and semantic properties (Haspelmath & Sims, 2013). Semantically, phrasal meanings such as "kick the bucket' or proper names like "Starbucks" (Figure 1) *exhibit emergent properties irreducible to their parts* (Partee, 1995). Vanilla SAEs demonstrably succeed at many interpretability tasks, yet their linear reconstruction mechanism cannot, in principle, represent non-linear composition. Without explicit interaction mechanisms, SAEs cannot simultaneously represent atomic features and their non-linear compositions. When "Starbucks" appears in context, a linear model must either (i) allocate a dedicated feature for this compositional entity, sacrificing atomicity, or (ii) represent it through separate "star" and "coffee" features that cannot distinguish this specific composition from mere co-occurrence.

Ideally, SAEs with sufficient capacity can learn features at

multiple levels of abstraction simultaneously (such as morphemes, words, phrases, and compositional expressions) co-existing as independent atoms in an overcomplete dictionary. While this leads to good reconstruction and intervention, it fundamentally limits our understanding: we cannot decompose "Starbucks" into its constituents, cannot trace how "administrators" emerge from stem and suffix binding, and cannot distinguish compositional phrases from accidental co-occurrence. The conflation of atomic and compositional features *obscures the mechanisms by which networks build complex representations from simpler parts*.

This problem connects to a *longstanding debate* in cognitive science about systematic compositionality in neural representations (Fodor & Pylyshyn, 1988). Smolensky (1990) proposed tensor product variable binding as a solution: *features bind through multilinear interactions* rather than linear superposition, allowing networks to maintain atomic constituents while representing their combinations. In this framework, "administrators" would be represented not as a single indivisible feature, but as an explicit *composition* of stem and suffix, where the tensor product captures the binding operation. For interpretability of modern LLMs, this principle is critical: to understand how networks compose meaning, our tools must themselves model compositional structure faithfully. However, explicit tensor products are computationally prohibitive for overcomplete sparse codes with tens of thousands of features, requiring methods that capture multilinear interactions while remaining tractable.

In this work, we introduce the **Polynomial Sparse Autoencoder (PolySAE)** (Figure 2), a sparse autoencoder that extends vanilla SAEs with explicit feature interaction terms. PolySAE preserves a linear encoder for interpretability while extending the decoder with quadratic and cubic terms that model pairwise and triple feature interactions. Through low-rank tensor factorization on a shared projection subspace, PolySAE captures compositional structure in

a tractable manner, adding a small parameter overhead (3% for GPT2 small). Critically, PolySAE is a strict *generalization of standard SAEs* that enables capturing multiplicative (non-additive) concept interactions. Setting interaction coefficients to zero recovers vanilla SAE behavior, allowing PolySAE to be readily applied to existing SAE variants, including TopK (Gao et al., 2025), BatchTopK (Bussmann et al., 2024), and Matryoshka (Bussmann et al., 2025).

We summarize below our **four main contributions**:

**C1.** We introduce **PolySAE**, a sparse autoencoder with a polynomial decoder that explicitly **models quadratic and cubic feature interactions** while preserving a linear encoder for interpretability. Through low-rank tensor factorization, PolySAE adds small parameter overhead (3% for GPT2 small) and can be **readily applied** to existing SAE variants (TopK, BatchTopK, Matryoshka).

**C2.** Across **four language models** of different scales (GPT-2 Small, Pythia-410M/1.4B, Gemma-2-2B) and **three sparsification strategies**, PolySAE achieves an **average 8% F1 improvement**, while maintaining comparable reconstruction error.

**C3.** PolySAE produces **2–10× larger Wasserstein distances** between class-conditional feature distributions, indicating more separated semantic structure in the learned representations.

**C4.** We show that learned interaction weights exhibit **negligible correlation with co-occurrence frequency** (r = 0.06 vs. r = 0.82 for SAE feature covariance), and provide qualitative examples demonstrating that **polynomial terms capture compositional structure** such as morphological binding, phrasal composition, and contextual disambiguation. We further show that the learned interaction directions causally steer model outputs toward the corresponding compositional semantics.

## 2. Related Work

**Sparse dictionary learning.** In sparse dictionary learning, signals are represented as sparse linear combinations of overcomplete basis elements (Mallat & Zhang, 1993), an approach also integrated into neural network architectures (Hinton & Salakhutdinov, 2006; Lee et al., 2007; Konda et al., 2014). Sparse Autoencoders (SAEs) recently emerged as a leading paradigm for feature discovery in large language models (Huben et al., 2024; Bricken et al., 2023), scaling to millions of features (Gao et al., 2025). Subsequent work has produced architectural variants including BatchTopK (Bussmann et al., 2024), Matryoshka (Bussmann et al., 2025), Gated (Rajamanoharan et al., 2024a), and JumpReLU (Rajamanoharan et al., 2024b) SAEs, with standardized benchmarks enabling systematic comparison (Kar-

vonen et al., 2025). However, all these methods assume features combine additively through linear reconstruction.

**Modeling feature interactions.** Multiplicative interactions between features have a rich history in deep learning (Jayakumar et al., 2020), from early bilinear models for visual data (Tenenbaum & Freeman, 1996; Freeman & Tenenbaum, 1997) to modern gating mechanisms (Shazeer, 2020). Feature interactions through the Hadamard product (Chrysos et al., 2025) serve as a powerful conditioning mechanism (Perez et al., 2018; Dumoulin et al., 2017), while multiplicative structure also enables parameter-efficient mixture-of-experts (Oldfield et al., 2025). Recent work has explored multiplicative interactions for interpretability: Bilinear MLPs (Pearce et al., 2025) model pairwise feature interactions enabling weight-based interpretability, while Gauderis & Dooms (2025) propose fully interpretable architectures based on tensor networks. We extend this line of work by modeling feature interactions in the SAE setting.

**Polynomials.** One natural way to model higher-order interactions is through polynomials (Shin & Ghosh, 1991). In deep learning, polynomials have been used for a variety of applications, such as image generation (Chrysos et al., 2020; 2021), classification (Babiloni et al., 2021; Chrysos et al., 2022a), privacy preservation (Zhang et al., 2019), interpretability (Dubey et al., 2022), and dynamic safety guardrails (Oldfield et al., 2026). The work most closely related to ours is the Bilinear Autoencoder (BAE) (Dooms & Gauderis, 2025), which similarly introduces interaction terms for interpretability. The key difference lies in the level at which interactions are modeled: BAE captures pairwise interactions between input neurons, whereas PolySAE models interactions directly between learned sparse features, including higher-order terms. As a result, PolySAE preserves the interpretability of linear SAE latents while explicitly allocating capacity to non-additive feature composition.

## 3. Sparse Polynomial Decoding

### 3.1. Preliminaries

**Notation.** Bold lowercase letters denote vectors and bold uppercase letters denote matrices. The $i$-th column of $M$ is $m_i$, and $M_{:,1:r}$ denotes its first $r$ columns. We use $*$ for the Hadamard product, $\otimes$ for the Kronecker product, and $\odot$ for the Khatri–Rao product. $\mathbb{R}^d$ and $\mathbb{R}^{d_{\text{sae}}}$ denote the activation and sparse-code spaces, with $d_{\text{sae}} \gg d$. $\mathcal{S}(\cdot)$ denotes a sparsification operator, such as Top-$K$ or BatchTop-$K$.

**Sparse Autoencoders.** Sparse autoencoders (SAEs) build on overcomplete dictionary learning (Olshausen & Field, 1997) to decompose neural activations into a sparse set of latent features. Given activations $x \in \mathbb{R}^d$ from an intermediate layer of a pretrained network, an SAE learns a sparse code $z \in \mathbb{R}^{d_{\text{sae}}}$ with $d_{\text{sae}} \gg d$ and reconstructs via

$\hat{x} = b_{\text{dec}} + D z, z = \mathcal{S}(\text{ReLU}(E^\top x + b_{\text{enc}}))$ where $E$ is a linear encoder, $D$ is the decoder (dictionary), and $\mathcal{S}$ enforces sparsity. The overcomplete latent space allows multiple features to align with similar activation directions, supporting disentangled and interpretable representations.

Motivated by the superposition hypothesis (Elhage et al., 2022), SAEs assume that features combine *additively* in the decoder, so reconstruction is linear in $z$. This corresponds to a strong form of the linear representation hypothesis applied to decoding, which has recently been questioned (Engels et al., 2025). When multiple features co-activate, their joint effect may not be well captured by a linear sum—for example, a "coffee" feature and a "star" feature may require a reconstruction direction distinct from either individual atom to capture the "Starbucks" concept. This motivates extending the decoder to explicitly model feature interactions, while preserving a linear and interpretable encoder.

### 3.2. Design Principles for Feature Interactions

We extend sparse autoencoders to capture higher-order feature interactions by establishing design principles grounded in prior work. Each architectural choice in PolySAE follows directly from these principles.

**P1. Linear Encoding** (Interpretability). Each sparse code coefficient $z_i$ is derived by a *linear projection* of the input activation $x$. The linear representation hypothesis in mechanistic interpretability posits that learned features should correspond to *directions* in activation space (Elhage et al., 2022; Bricken et al., 2023), a view supported by the success of linear probes for extracting semantic content (Belinkov, 2022; Alain & Bengio, 2016).

**P2. Polynomial Reconstruction** (Expressivity). The decoder may capture compositional structure by using polynomial terms in $z$. Modeling *compositional* structure, *i.e.* how features interact, polynomials have a strong precedent in the literature: Volterra series (Volterra, 1959) represent nonlinear systems as sums of multilinear kernels, second-order pooling (Carreira et al., 2012; Gao et al., 2016) captures feature co-occurrences via outer products, and polynomial networks (Chrysos et al., 2022b) parameterize functions as products of linear projections.

**P3. Factorized Interaction Structure** (Coherence & Efficiency). Higher-order terms should operate in a low-dimensional subspace aligned with the linear feature space. Using a shared projection $U$ ensures that interactions are compositions of the same underlying features. This alignment principle underlies factorized interaction models (Rendle, 2010; Blondel et al., 2016) and compact bilinear pooling (Gao et al., 2016; Kim et al., 2017). Constraining interactions to low-rank subspaces imposes a strong inductive bias, favoring coherent, reusable interaction modes over arbitrary pairwise composition.

**P4. Structural Constraints** (Parsimony & Identifiability). Lower-order terms should have higher representational capacity than higher-order terms, following polynomial approximation theory (Mason & Handscomb, 2002). The latent interaction subspace should have orthonormal columns to ensure geometrically distinct directions. Orthogonality constraints are standard in dictionary learning and independent component analysis to prevent degenerate solutions (Arora et al., 2015; Bao et al., 2016; Hyvärinen & Oja, 2000). Orthonormality removes rotational ambiguity and ensures the model does not allocate redundant capacity to correlated interaction directions.

### 3.3. PolySAE: Polynomial Sparse Autoencoder

To satisfy **P1** PolySAE adopts the standard SAE encoder (Huben et al., 2024; Bricken et al., 2023) to first performs a linear map followed by sparsification:

$$z = \mathcal{S}(\text{ReLU}(E^\top x + b_{\text{enc}})), \qquad z \in \mathbb{R}^{d_{\text{sae}}}, \quad (1)$$

where feature $i$ activates when $x$ aligns with direction $e_i$, enabling visualization, clustering, and causal intervention via activation patching (Meng et al., 2022).

Following **P2**, we extend the decoder to include quadratic and cubic terms:

$$\hat{x} = b_{\text{dec}} + y_1 + \lambda_2\, y_2 + \lambda_3\, y_3, \qquad (2)$$

where $y_1 = A\,z$, $y_2 = B\,(z \otimes z)$, $y_3 = \Gamma\,(z \otimes z \otimes z)$, and $\lambda_2, \lambda_3 \in \mathbb{R}$ are learnable scalar coefficients that control the contribution of each polynomial order. Setting $\lambda_2 = \lambda_3 = 0$ recovers a standard linear sparse autoencoder, making PolySAE a strict generalization of existing SAE architectures. This can be viewed as a third-order Volterra expansion (Volterra, 1959) or a $\Pi$-*net* polynomial parameterization (Chrysos et al., 2022b), adapted to sparse codes.

However, explicitly modeling all pairwise or higher-order feature combinations would require $O(d_{\text{sae}}^2)$ or $O(d_{\text{sae}}^3)$ parameters, leading to unstructured interaction effects and a high risk of overfitting. Following **P3** and **P4**, we constrain interactions to a low-rank subspace:

$$y_1 = (z\,U)\,C^{(1)\top},$$

$$y_2 = ((z\,U_{:,1:R_2}) * (z\,U_{:,1:R_2}))\,C^{(2)\top},$$

$$y_3 = ((z\,U_{:,1:R_3}) * (z\,U_{:,1:R_3}) * (z\,U_{:,1:R_3}))\,C^{(3)\top},$$
$$(3)$$

where $*$ denotes element-wise product and $C^{(k)} \in \mathbb{R}^{d \times R_k}$ are output projection matrices. This parameterization restricts the interaction dictionaries to rank at most $R_k$, enforcing a strong inductive bias on how features may combine.

Notice that this parameterization satisfies **P3** by applying a single projection $U$ to the sparse code and forming interactions via polynomial operations: $zU$, $(zU) * (zU)$, and $(zU) * (zU) * (zU)$. Using the same projected representation at every order ensures that interaction effects remain aligned with the linear feature basis and interpretable as compositions of the same underlying features.

Furthermore, PolySAE satisfies the parsimony aspect of **P4** by following nested low-rank approximation (Grasedyck et al., 2013) and utilizing ranks $(R_1, R_2, R_3)$ with $R_1 \geq R_2 \geq R_3$. This nested structure means $\mathrm{span}(U_{:,1:R_3}) \subset \mathrm{span}(U_{:,1:R_2}) \subset \mathrm{span}(U)$. In practice, $R_2 = R_3 \ll R_1$ (*e.g.*, $R_2 = R_3 = 64$) suffices to capture most interaction structure, confirming our hypothesis that higher-order contributions are low-dimensional (Section 4).

Finally, to satisfy the identifiability aspect of **P4**, we enforce orthonormality of the interaction subspace. Following Stiefel optimization (Absil et al., 2008; Bonnabel, 2013), we impose $U^\top U = I$ via QR retraction after each gradient step. We use positive QR retraction (Edelman et al., 1998), which corrects column signs to ensure continuity and avoids discontinuous representation changes during training.

### 3.4. Discussion

**Context-Dependent Dictionary Structure.** In standard SAEs, each feature $i$ is associated with a fixed dictionary atom $d_i$: regardless of context, activating feature $i$ contributes $z_i d_i$ to the reconstruction. PolySAE fundamentally alters this picture. Because reconstruction includes higher-order terms, the effective contribution of a feature becomes *context-dependent*, varying with which other features are simultaneously active.

This can be seen by expanding Equation (2). The linear term defines a dictionary over individual features, while the quadratic and cubic terms define dictionaries over feature pairs and triples, respectively. Under our low-rank factorization, these dictionaries are implicitly given by

$$A = C^{(1)} U^\top \qquad\qquad \in \mathbb{R}^{d \times d_{\mathrm{sae}}},$$

$$B = C^{(2)}(U_{:,1:R_2} \odot U_{:,1:R_2})^\top \qquad \in \mathbb{R}^{d \times d_{\mathrm{sae}}^2},$$

$$\Gamma = C^{(3)}(U_{:,1:R_3} \odot U_{:,1:R_3} \odot U_{:,1:R_3})^\top \quad \in \mathbb{R}^{d \times d_{\mathrm{sae}}^3}, \tag{4}$$

where $A$ is the *linear dictionary*, $B$ the *pairwise interaction dictionary*, and $\Gamma$ the *triple interaction dictionary*. Column $(i, j)$ of $B$ specifies how the co-activation $z_i z_j$ modifies the reconstruction, while column $(i, j, k)$ of $\Gamma$ specifies the contribution arising from the joint activation $z_i z_j z_k$. The computational form in Equation (3) is algebraically equivalent to Equation (4) but avoids explicitly materializing the $d_{\mathrm{sae}}^2$- and $d_{\mathrm{sae}}^3$-dimensional dictionaries.

**Compositional Capacity.** Using the same $d_{\mathrm{sae}}$ base features as an SAE, PolySAE can support interaction-driven structure across $\binom{d_{\mathrm{sae}}}{2} \cdot R_2 + \binom{d_{\mathrm{sae}}}{3} \cdot R_3$ feature pairs and triples, enabling a substantially larger space of distinct semantic compositions without increasing the number of features. This capacity is mediated through a shared low-rank interaction space: rather than allocating independent parameters to each feature combination, interactions are expressed via $R_2$ and $R_3$ shared modes. As a result, potential feature combinations are realized through a small number of reusable interaction directions, reflecting the empirically observed low-dimensional structure of feature interactions.

**Parameter Efficiency.** PolySAE modifies only the decoder; the encoder is unchanged. A standard SAE has $2d\, d_{\mathrm{sae}} + d + d_{\mathrm{sae}}$ parameters. When the linear term is full rank ($R_1 = d$), PolySAE adds $\Delta P = d^2 + d(R_2 + R_3) + 2$ parameters. With the empirically optimal choice $R_2 = R_3$ and $R_2 \in [0.06R_1, 0.11R_1]$, this yields $\Delta P = (1.12\text{–}1.22)\, d^2$ (up to constants). For GPT-2 small ($d = 768$, $d_{\mathrm{sae}} = 16{,}384$), this corresponds to an increase of $\sim 2.5\text{–}3\%$ of the full SAE.

## 4. Empirical Evaluation

### 4.1. Experimental Setup

Our training pipeline is built by extending `SAELens` (Bloom et al., 2024) to include PolySAE. We train and evaluate our methods against the standard SAE with **three sparsification strategies**, TopK (Gao et al., 2025), BatchTopK (Bussmann et al., 2024), and Matryoshka (Bussmann et al., 2025). Throughout all experiments, we use a sparsity level of $K = 64$ with 16,384 latents trained on residual-stream activations from **four pretrained language models of different scales**: Gemma-2-2B (Gemma Team, 2024) (layer 19), Pythia-410M and Pythia-1.4B (Biderman et al., 2023) (layers 15 and 12, respectively), and GPT-2 Small (Radford et al., 2019) (layer 8). Training uses 500M tokens (300M for GPT-2 Small) with context length 128. For Gemma-2-2B and GPT-2 Small, we use OpenWebText (Gokaslan et al., 2019); for Pythia models, we use an uncopyrighted variant of the deduplicated Pile (Gao et al., 2021). We evaluate learned features using SAEBench (Karvonen et al., 2025), which reports reconstruction metrics on held-out data from the training distribution and sparse probing performance on **six classification tasks**: Bias in Bios (De-Arteaga et al., 2019), AG News (Zhang et al., 2015), EuroParl (Koehn, 2005), GitHub programming languages (CodeParrot, 2022), Amazon Sentiment, and Amazon-15 (Hou et al., 2024). For more implementation details see Section B.

### 4.2. Reconstruction and Semantic Modeling

We evaluate models along two axes: *(Q1) reconstruction fidelity* and *(Q2) semantic modeling of the learned representations*. Reconstruction quality is measured using mean

*Table 1.* F1 Scores (%) across datasets at K=1. Format: F1 / Wasserstein ($\times 10^{-3}$). Mean Probing column shows mean F1 across datasets. MSE for reconstruction error and CE Rec. denotes cross-entropy recovery.

| LLM | SAE variant | MSE | CE Rec. | Mean F1 | Europarl | Bios | Amazon Sentiment | GitHub | AG News | Amazon 15 |
|---|---|---|---|---|---|---|---|---|---|---|
| GPT-2 Small | Topk | **0.52** | **0.993** | 67.1 | 67.7 / 19.0 | 61.0 / 7.7 | 76.0 / 4.3 | 63.4 / 8.7 | 71.4 / 8.5 | 63.3 / 2.8 |
| | Topk + PolySAE | 0.55 | **0.993** | **77.9** | **86.1 / 35.2** | **75.5 / 16.8** | **83.1 / 9.7** | **73.0 / 20.6** | **81.0 / 18.9** | **69.0 / 6.7** |
| | BTopk | **0.53** | **0.993** | 65.7 | 67.4 / 19.0 | 59.6 / 7.3 | 68.8 / 4.4 | 68.1 / 8.8 | 65.3 / 8.2 | **65.1 / 2.9** |
| | BTopk + PolySAE | 0.54 | **0.993** | **78.0** | **92.0 / 39.9** | **70.9 / 17.3** | **84.2 / 8.5** | **74.4 / 18.5** | **83.2 / 20.0** | 63.2 / 6.0 |
| | Matryoshka | 0.60 | **0.992** | 65.7 | 65.8 / 12.5 | 61.2 / 4.0 | 76.2 / 3.2 | 60.9 / 7.9 | 68.1 / 4.3 | 62.1 / 2.4 |
| | Matr. + PolySAE | **0.58** | **0.992** | **77.7** | **95.0 / 30.0** | **72.9 / 14.3** | **81.4 / 8.1** | **71.5 / 18.6** | **77.4 / 16.0** | **68.0 / 5.6** |
| Pythia-410m | Topk | **0.03** | 0.971 | 71.2 | 96.1 / 2.0 | 67.4 / 1.1 | 61.5 / 0.7 | 64.6 / 1.5 | 71.8 / 1.2 | 65.9 / 0.4 |
| | Topk + PolySAE | 0.04 | 0.970 | **77.0** | **96.7 / 6.8** | **70.8 / 3.8** | **75.9 / 2.3** | **74.0 / 5.3** | **73.3 / 4.0** | **71.5 / 1.4** |
| | BTopk | **0.03** | 0.973 | 65.0 | 90.9 / 0.8 | 60.5 / 0.3 | 63.9 / 0.4 | 59.7 / 1.1 | 58.7 / 0.3 | 56.6 / 0.3 |
| | BTopk + PolySAE | 0.04 | 0.971 | **77.3** | **97.8 / 8.2** | **74.0 / 4.1** | **74.6 / 2.1** | **75.2 / 4.8** | **78.6 / 4.4** | **63.5 / 1.3** |
| | Matryoshka | **0.04** | 0.969 | 64.2 | 79.1 / 0.6 | 63.6 / 0.3 | 64.4 / 0.4 | 62.3 / 1.1 | 58.7 / 0.3 | 57.3 / 0.3 |
| | Matr. + PolySAE | **0.04** | 0.972 | **74.6** | **99.2 / 2.8** | **71.0 / 1.2** | **66.9 / 1.3** | **81.8 / 3.7** | **64.8 / 1.2** | **63.8 / 0.9** |
| Pythia-1.4b | Topk | **0.23** | 0.971 | 75.9 | **97.8 / 1.6** | 72.4 / 1.4 | 69.5 / 0.8 | 69.3 / 1.9 | 77.3 / 1.4 | 69.0 / 0.5 |
| | Topk + PolySAE | **0.23** | 0.973 | **81.9** | 96.8 / 7.9 | **77.2 / 6.4** | **88.1 / 3.7** | **74.7 / 9.2** | **83.4 / 6.3** | **71.1 / 2.4** |
| | BTopk | **0.22** | **0.975** | 64.6 | 74.0 / 0.6 | 65.0 / 0.5 | 57.2 / 0.6 | 63.3 / 2.3 | 65.2 / 0.4 | 63.2 / 0.4 |
| | BTopk + PolySAE | 0.23 | 0.974 | **76.4** | **93.7 / 4.5** | **73.0 / 3.4** | **67.1 / 3.1** | **73.8 / 8.2** | **77.6 / 3.4** | **73.2 / 2.1** |
| | Matryoshka | 0.24 | 0.970 | 64.4 | 70.2 / 0.5 | 62.8 / 0.5 | **63.1 / 0.6** | 65.6 / 1.9 | 64.1 / 0.4 | 60.8 / 0.4 |
| | Matr. + PolySAE | **0.23** | 0.973 | **72.1** | **91.1 / 2.9** | **72.4 / 2.0** | 58.0 / 2.1 | **68.2 / 6.7** | **73.6 / 2.1** | **69.4 / 1.5** |
| Gemma2-2b | Topk | **1.59** | **0.988** | 67.7 | 78.6 / 5.3 | **69.6 / 7.1** | **71.8 / 4.4** | 60.7 / 6.1 | 60.7 / 7.2 | 64.8 / 2.7 |
| | Topk + PolySAE | 1.65 | 0.987 | **68.4** | **86.8 / 12.0** | 64.7 / 16.8 | 64.5 / 10.5 | **64.1 / 16.1** | **61.9 / 16.9** | **68.5 / 6.3** |
| | BTopk | **1.58** | **0.988** | 64.8 | 68.3 / 1.9 | 67.6 / 2.6 | **71.1 / 2.8** | 64.4 / 4.5 | 59.9 / 2.7 | 57.6 / 1.9 |
| | BTopk + PolySAE | 1.68 | 0.987 | **69.4** | **92.8 / 13.2** | **78.3 / 18.3** | 56.4 / 10.2 | **65.0 / 16.1** | **64.0 / 18.8** | **60.0 / 6.4** |
| | Matryoshka | 1.69 | **0.987** | 60.9 | 60.8 / 0.7 | 64.0 / 0.8 | 57.3 / 1.5 | 61.5 / 2.5 | **61.0 / 0.8** | 60.5 / 1.0 |
| | Matr. + PolySAE | **1.64** | **0.987** | **65.6** | **77.6 / 2.1** | **67.5 / 3.1** | **61.7 / 4.9** | **63.7 / 8.8** | 60.9 / 3.5 | **62.2 / 3.3** |

squared error between the decoder output and the unnormalized network activations. To assess semantic structure, we use two complementary metrics.

*Probing.* We evaluate the linear separability of semantic concepts in the learned sparse representations by training logistic regression classifiers on SAE activations to predict ground-truth labels across multiple datasets. For each task, classification is performed using the feature with the largest mean activation difference between positive and negative classes, isolating semantic signal at the feature level.

*Distributional separation.* Probing relies on post-hoc decision boundaries and may not fully reflect the intrinsic geometry of the representation. We therefore additionally compute the 1-Wasserstein distance between class-conditional activation distributions. Unlike probing, which evaluates separability at a specific threshold, the Wasserstein distance captures global distributional separation, with larger values indicating more distinct semantic separation across space.

Table 1 demonstrates that **across four language models and three sparsification strategies**, PolySAE achieves *comparable MSE to standard SAE* across all configurations, confirming that polynomial decoding does not sacrifice reconstruction fidelity. For probing, PolySAE **consistently outperforms SAE by large margins** with mean gains of more than 10% on GPT-2, and 8% on average across models (Pythia-410M, Pythia-1.4B, and Gemma2-2B) and sparsi-

fiers. Crucially, PolySAE also achieves **consistently substantially higher Wasserstein distances**, with improvements of approximately 2–10$\times$ across all other models. This indicates that the gains observed in probing accuracy reflect *genuinely better-separated class-conditional representations*, rather than improvements driven solely by favorable decision boundaries. We additionally report cross-entropy (CE) loss recovery in Table 1, which captures whether the SAE preserves model behavior when its reconstruction substitutes for the original activation. Across all 12 configurations PolySAE's CE recovery lies within 0.003 of vanilla SAE, confirming that the small MSE differences do not translate into functional degradation.

### 4.3. Ablations

**Ablating P3 and P4.** We next ask *(Q3) whether each architectural choice in PolySAE carries its weight*. Starting from a polynomial decoder with full-rank shared projections, we additively introduce low-rank factorization (P3) and orthogonality (P4). Table 2 reports parameters, MSE, and F1 on GPT-2 Small. Low-rank factorization cuts parameters by *65% at only 1pp F1 cost*, making training tractable. Orthogonality then *recovers the gap and surpasses the original polynomial by +2.9pp at zero parameter cost*, validating P4.

**PolySAE Enables Competitive Performance with Sparser Codes.** We ask *(Q4) whether PolySAE's capacity to model feature interactions enables the use of sparser rep-

*Table 2.* Architectural ablation on GPT-2 Small validating P3 and P4.

| Ablation | Params | MSE | F1 |
|---|---|---|---|
| Polynomial + shared proj. | 37.7M | 0.58 | 76.0 |
| + low-rank factorization | 13.3M | **0.53** | 75.0 |
| + orthogonality (PolySAE) | 13.3M | 0.55 | **77.9** |

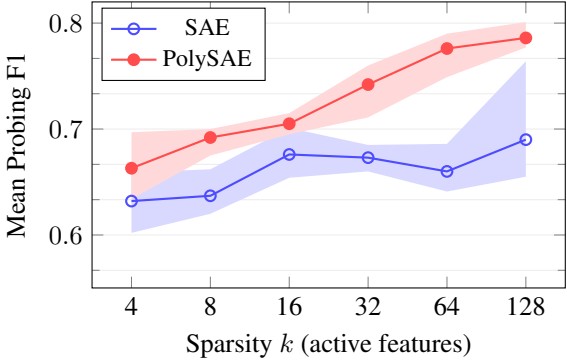

*Figure 3.* **Probing Mean F1 vs. sparsity** $k$. Shaded regions show range across widths (2k–16k). PolySAE consistently outperforms SAE with significant separation at higher $k$.

*Table 3.* Mean F1 Gain from K=1 to K=5 per model and sparsifier, averaged across the 6 probing datasets. **Bold** marks the lower (better) value per row, indicating stronger semantic concentration into fewer features.

| LLM | Sparsifier | SAE | PolySAE |
|---|---|---|---|
| GPT-2 Small | TopK | +15.2 | **+6.8** |
| GPT-2 Small | BatchTopK | +13.6 | **+6.5** |
| GPT-2 Small | Matryoshka | +14.2 | **+6.8** |
| Pythia-410m | TopK | +10.2 | **+9.8** |
| Pythia-410m | BatchTopK | +12.1 | **+11.2** |
| Pythia-410m | Matryoshka | +10.7 | **+10.4** |
| Pythia-1.4b | TopK | +10.2 | **+6.6** |
| Pythia-1.4b | BatchTopK | **+9.7** | +10.6 |
| Pythia-1.4b | Matryoshka | **+11.1** | +14.3 |
| Gemma2-2b | TopK | +16.9 | **+9.2** |
| Gemma2-2b | BatchTopK | **+12.6** | +12.8 |
| Gemma2-2b | Matryoshka | +11.2 | **+9.7** |

*resentations*. Figure 3 shows probing F1 as a function of active features $k$, with shaded regions indicating variance across dictionary widths (2k–16k). PolySAE consistently outperforms standard SAEs across all sparsity levels, with the gap widening at higher $k$. PolySAE also exhibits lower variance across widths, enabling competitive performance with fewer active features.

**Semantic Concentration Across Features.** We further ask *(Q5) whether PolySAE concentrates semantic signal into fewer features*. Table 3 reports the F1 gain $\Delta_{1-5}$ when expanding from K=1 to K=5 active features, broken down by model and sparsifier across the probing datasets. PolySAE exhibits smaller gains than standard SAEs in 9 out of 12 model×sparsifier configurations, with all three GPT-2 Small sparsifiers showing gaps of $-7$ to $-8$. This behavior is probably due to the fact that higher-order interactions absorb contextual variability while PolySAE's linear features remain more semantically focused.

## 5. Understanding and Utilizing Interactions

### 5.1. Making Sense of Learned Interactions

To better understand the learnt interactions, we firstly ask *(Q6) whether PolySAE's higher-order terms encode genuine compositional structure or merely reflect surface-level co-occurrence*. To study this, we analyze SAE and PolySAE activations trained with Top-$K$ sparsification on GPT-2 small, using 1M OpenWebText texts. For each feature pair $(i, j)$, we first define the learned quadratic interaction strength $B_{ij} = \lambda_2 \left\| (\boldsymbol{u}_i \odot \boldsymbol{u}_j)^\top \boldsymbol{C}^{(2)} \right\|_2$, which depends only on the trained decoder parameters and measures how much representational capacity the model allocates to the $(i, j)$ interaction. Second, we compute the empirical co-occurrence frequency $N_{ij}$ by counting token positions in which both features appear in the top-$K$ active set across the same corpus. If polynomial interactions merely replicated bigram statistics, $B_{ij}$ and $N_{ij}$ would correlate strongly. As a baseline, we consider the empirical covariance of SAE activations, which captures the full pairwise structure accessible to a linear model.

As expected, this covariance correlates strongly with co-occurrence frequency ($r = 0.82$). In contrast, PolySAE's learned interactions exhibit negligible correlation with co-occurrence ($r = 0.06$), indicating that interaction capacity is allocated based on criteria largely orthogonal to frequency.

Since higher-order dictionaries do not simply encode co-occurrence, we ask *(Q7) whether the learned interactions are interpretable*. To analyze this structure, we construct a dictionary mapping each feature to its most activating tokens, then examine feature pairs and triples with high interaction strength by extracting representative contexts in which the corresponding features co-activate.

Selected examples in Table 4 and Table 5 illustrate the qualitative structure captured by PolySAE's higher-order terms. Second-order interactions often correspond to coherent phrase-level compositions that are not recoverable from either feature in isolation, such as *coffee × star* yielding contexts referring to *Starbucks*, a highly non-linear semantic mapping. In contrast, SAEs typically activate broad or weakly related features in these contexts, failing to recover the composed meaning. Third-order interactions further

*Table 4.* **Second-Order Interaction Examples Captured by PolySAE.** Quadratic interactions bind two features to capture context-dependent semantic structure beyond co-occurrence. SAE often recovers individual components but fails to represent the composed meaning.

| Poly $F_1$ | Poly $F_2$ | Context | SAE | Observed Pattern |
|---|---|---|---|---|
| [star, stars] | [coffee, tea] | We've all certainly heard of beers brewed with espresso, but how about one with an espresso shot poured over the top? **Starbucks** | [Apple, Google] | *The interaction binds features to represent a specific named entity creating a new semantic category.* |
| [surgery, repair] | [Trans, LGBT] | Some in the transgender community are worried a suspicious fire at a Montreal clinic will add delays to an already lengthy process to get gender reassignment **surgery** | [birth, baby] | *Specialization: a general concept (surgery) gets specialized by domain context (Trans,LGBT) narrowing the semantic scope.* |
| [DNA, genetic] | [mod, mods] | Activists are opening up a new front in their campaign against genetic modification. The latest target is genetically-**mod**ified trees | [modified, edit] | *Multiple modifiers stack to create specific compound meanings. Interaction binds genetic with the action modification.* |
| [secret, hidden] | [Snowden, WikiLeaks] | On May 24th PBS aired a Frontline documentary about alleged Wikileaker Bradley Manning called "**WikiSecrets**" | [secret, secrets] | *Feature interaction binds topical concepts to create coined term that could not be modeled via co-occurrence alone.* |

*Table 5.* **Third-Order Interaction Examples Captured by PolySAE.** Cubic interactions condition pairwise compositions on additional context, disambiguating meaning through three-way binding. Vanilla SAE typically activates broader or less specific features.

| Poly $F_1$ | Poly $F_2$ | Poly $F_3$ | Context | SAE | Observed pattern |
|---|---|---|---|---|---|
| [proved, proven] | [star, stars, superstar] | [reputation, fame] | David Bowie proved some **stars** are big enough not to make themselves available | [star, stars, superstar] | *Three-way relational binding, all arguments must be present; reputation disambiguates which aspect of stars is relevant to the proving action.* |
| [nuclear, reactor] | [test, testing] | [radiation, magnetic] | US tests **nuclear**-capable missile with the range to strike North Korea | [nuclear, atomic] | *Specifying concept; Event type (testing) × domain (nuclear) × capability (radiation) (Parsons, 1990)* |
| [black, racial] | [Americans, Canadians] | [people, women] | In a push to get more Black **Americans** involved in the world of tech | [Americans, Muslims, Jews] | *Multi-attribute category intersection, binding demographic attributes.* |
| [ing, ting] | [stock, market] | [invest, investing] | **Invest**ing.com — Philippines stocks were higher after | [ing, training, running] | *Three-way interaction between morphological marker (ing) and domain (stock, market) (Asher, 2011).* |

refine such compositions by conditioning on additional context. For example, PolySAE distinguishes financial *investing* from unrelated -ing usages by integrating morphological cues with market-related features, and disambiguates generic entities such as *nuclear* or *Americans* based on surrounding semantic attributes. Across examples, **higher-order terms absorb contextual variation that would otherwise fragment linear features**, allowing PolySAE to express compositional meaning through structured interactions rather than proliferating context-specific atoms. Further examples in Section C confirm these patterns.

**Quantifying interpretability at scale.** The qualitative examples above show that selected interactions are interpretable; we now ask *(Q8) what fraction of PolySAE's learned interactions are interpretable*. We query GPT-4o-mini with a structured prompt that returns a 0–1 score for

how strongly each interaction matches the composite concept implied by its two constituent features (Section D). Of the $\sim 5 \times 10^7$ candidate pairs over the top 10K features by activation mass, 292,361 exhibit non-negligible interaction strength $B_{ij}$. Of the 70,000 pairs we evaluated, 8,550 score above 0.9, a 12% rate of highly interpretable compositional interactions. Even assuming not every one of the 16K linear features is itself interpretable, *PolySAE adds at least 8,550 compositional concepts at the second-order level alone on top of the linear dictionary*.

### 5.2. Steering with Learned Interactions

Having established that the learned interactions are interpretable; we now ask *(Q9) whether the directions they define are causally useful for steering model behavior*. For each of 27 compositional concepts drawn from Table 4 and Sec-

*Table 6.* Selected examples of activation steering on GPT-2 Small. We add the first-order decoder directions $\mathbf{d}_i + \mathbf{d}_j$ to the residual stream (layer 8) during greedy generation. Each row shows the continuation under three conditions: no steering, vanilla SAE, and PolySAE. **Bold** marks where PolySAE steers generation toward the compositional target. The full set of examples is in Table 14.

| Features → Target | Prompt | No Steering | SAE | PolySAE |
|---|---|---|---|---|
| [surgery]×[trans] → *gender* | "The procedure that helps individuals align their body with their identity is" | called "*body alignment*." The procedure involves the body aligning. . . | called "*body alignment*." The procedure involves the use of a combination. . . | called "***gender identity surgery***." Performed by a surgeon who specializes in gender. . . |
| [canada]×[oil] → *Keystone* | "The controversial cross-border pipeline project is called the" | *Trans-Pacific Partnership* (TPP), a major step forward for the U.S. and Canada. | *Trans Mountain pipeline*, a controversial project in the works for years. | ***Keystone XL***. The pipeline would carry crude oil from Alberta to U.S. refineries. |
| [involved]×[support] → *community* | "The foundation wants more people to become" | *entrepreneurs*, and it wants to make sure they're not just part of the problem. | *entrepreneurs*, and it wants to make sure they're not just part of the problem. | ***involved in the community***, and to help them make a difference. |
| [economic]×[times] → *Economist* | "The magazine with coverage of world politics and business is The" | *New York Times*. The magazine with coverage of world politics and business is The New York Times. | *New York Times*. The New York Times is a daily newspaper in the United States. . . | ***Economist***. The Economist is a magazine that is a global news magazine. . . |

tion C, we form a steering vector $d_i + d_j$ from the two corresponding decoder directions and inject $\alpha(d_i + d_j)$ into GPT-2's layer-8 residual stream at every token position during autoregressive generation (50 tokens). For each concept we design 12 neutral prompts in which the compositional target could plausibly appear (e.g., *"The controversial cross-border pipeline project is called the"*), giving 324 prompt-concept pairs, and compare PolySAE directions against vanilla SAE directions and no steering. Each condition is evaluated by the rank of the target compositional token in the model's next-token distribution; lower rank is better.

**Steering with compositional directions.** PolySAE *shifts generation toward the compositional target far more reliably* than vanilla SAE. Across all 324 prompt-concept pairs, PolySAE achieves lower target rank than no steering in 230 cases (71.0%) with only 3 degradations. Aggregating to the concept level, PolySAE improves over no steering in 27/27 concepts and over vanilla SAE in 21/27, with a mean rank improvement of +41.5 against vanilla. Representative examples appear in Table 6: steering with [canada]×[oil] produces *Keystone XL* under PolySAE, while vanilla produces *Trans Mountain pipeline* and no steering defaults to the unrelated *Trans-Pacific Partnership*. Across concepts, vanilla SAE often fails to shift output away from the unsteered baseline at all, while PolySAE recovers the intended composition.

**Direction alignment.** We verify that these gains reflect better-aligned representations. For each concept, we estimate a ground-truth compositional direction using the difference-in-means method (Marks & Tegmark, 2024),

which AxBench (Wu et al., 2025) demonstrated to be among the most effective approaches for identifying causally efficacious concept directions. We collect 20 sentences containing the concept and 20 without, extract GPT-2 layer-8 activations at the target token, and compute $\hat{d}_{\text{concept}} = \text{normalize}(\bar{a}_{\text{pos}} - \bar{a}_{\text{neg}})$, where $\bar{a}_{\text{pos}}$ and $\bar{a}_{\text{neg}}$ are the mean activations across the positive and negative sets. We then measure cosine similarity between each model's $d_i + d_j$ and this ground truth. PolySAE achieves mean cosine similarity $0.372 \pm 0.093$ vs. $0.311 \pm 0.158$ for vanilla SAE, a 19.7% relative improvement (11/27 wins vs. 2/27, 14 ties).

## 6. Conclusion

We introduced PolySAE, a sparse autoencoder that extends the decoder with higher-order terms to model feature interactions while preserving a linear encoder for interpretability. Through low-rank tensor factorization on a shared projection subspace, PolySAE captures pairwise and triple interactions with small parameter overhead. Across four LLMs and three SAE variants, it improves probing F1 by 8% on average while maintaining comparable reconstruction, achieves 2–10× larger Wasserstein distances between class-conditional distributions, and allocates interaction capacity based on compositional structure rather than co-occurrence ($r = 0.06$). The learned interaction directions causally steer model outputs toward the corresponding compositional semantics. **Limitations:** we study models up to 2B parameters and restrict experiments to forced-sparsity SAE variants.

## Acknowledgements

This work is supported by the TensorICE project (EXCEL-LENCE/0524/0407), implemented under the social cohesion programme "THALIA 2021-2027", co-funded by the European Union through the Research and Innovation Foundation. Yannis Panagakis was supported by the project MIS 5154714 of the National Recovery and Resilience Plan Greece 2.0 funded by the European Union under the NextGenerationEU Program. The authors gratefully acknowledge the EuroHPC Joint Undertaking for awarding this project access to the MareNostrum5 supercomputer, hosted by the Barcelona Supercomputing Center, Spain, under Development Access project ID EHPC-DEV-2025D06-069. This work was also supported by computing time awarded on the Cyclone supercomputer of the High Performance Computing Facility of The Cyprus Institute.

## Impact Statement

This work contributes to the field of interpretable machine learning by introducing PolySAE, a method for modeling non-additive feature interactions in sparse autoencoders. By enabling explicit representation of compositional structure while preserving linear, human-interpretable features, this approach advances tools for mechanistic analysis of large language models. Improved interpretability has the potential to support downstream efforts in model auditing, debugging, and safety research by making it easier to identify, analyze, and intervene on meaningful internal representations.

The primary anticipated benefits of this work are methodological and scientific. PolySAE is intended as an analysis tool rather than a deployment-facing component, and it does not directly increase the capabilities of language models. As with other interpretability methods, there is a possibility that insights into internal representations could be misused to more effectively manipulate model behavior, but we do not identify novel or unique risks introduced by this work beyond those already present in the interpretability literature.

Overall, we believe this work has a net positive societal impact by strengthening the technical foundations of interpretability and contributing to the long-term goal of building more transparent, controllable, and trustworthy machine learning systems.

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

---

**Algorithm 1** PolySAE Training

---

**Input:** activations $\{\boldsymbol{x}\}$, ranks $(R_1, R_2, R_3)$, sparsity $K$, learning rate $\eta$, rescale by decoder norm flag $\rho$

Initialize $\boldsymbol{U} \leftarrow \mathrm{qr}_+(\boldsymbol{U}_{\mathrm{rand}}); \quad \lambda_2 \leftarrow -0.5; \quad \lambda_3 \leftarrow 0.5$

**for** each minibatch $\boldsymbol{x}$ **do**

    *// Encode*

    $\boldsymbol{h} \leftarrow \boldsymbol{E}^\top \boldsymbol{x} + \boldsymbol{b}_{\mathrm{enc}}$

    **if** $\rho$ **then**

        Compute decoder norms $\boldsymbol{d} \in \mathbb{R}^{d_{\mathrm{sae}}}: d_i = \|\mathrm{PolyDec}(\boldsymbol{e}_i)\|_2$

        $\boldsymbol{z} \leftarrow \mathrm{TopK}(\mathrm{ReLU}(\boldsymbol{h} \odot \boldsymbol{d}), K)$

        $\boldsymbol{z} \leftarrow \boldsymbol{z}/\boldsymbol{d}$      *// rescale back for decode invariance*

    **else**

        $\boldsymbol{z} \leftarrow \mathrm{TopK}(\mathrm{ReLU}(\boldsymbol{h}), K)$

    **end if**

    *// Decode (polynomial, shared, hierarchical)*

    $\boldsymbol{y}_1 \leftarrow (\boldsymbol{z}\,\boldsymbol{U})\,\boldsymbol{C}^{(1)\top}$

    $\boldsymbol{A}_2 \leftarrow \boldsymbol{z}\,\boldsymbol{U}_{:,1:R_2}$

    $\boldsymbol{y}_2 \leftarrow (\boldsymbol{A}_2 * \boldsymbol{A}_2)\,\boldsymbol{C}^{(2)\top}$

    $\boldsymbol{A}_3 \leftarrow \boldsymbol{z}\,\boldsymbol{U}_{:,1:R_3}$

    $\boldsymbol{y}_3 \leftarrow (\boldsymbol{A}_3 * \boldsymbol{A}_3 * \boldsymbol{A}_3)\,\boldsymbol{C}^{(3)\top}$

    $\boldsymbol{y} \leftarrow \boldsymbol{b}_{\mathrm{dec}} + \boldsymbol{y}_1 + \lambda_2\,\boldsymbol{y}_2 + \lambda_3\,\boldsymbol{y}_3$

    *// Update with manifold retraction*

    $\mathcal{L} \leftarrow \|\boldsymbol{y} - \boldsymbol{x}\|_2^2 + \text{regularizations}$

    Update all parameters via $\nabla\mathcal{L}$

    $(\boldsymbol{Q}, \boldsymbol{R}) \leftarrow \mathrm{qr}(\boldsymbol{U}); \quad \boldsymbol{S} \leftarrow \mathrm{diag}(\mathrm{sgn}(\mathrm{diag}(\boldsymbol{R}))); \quad \boldsymbol{U} \leftarrow \boldsymbol{Q}\boldsymbol{S}$

**end for**

---

# A. PolySAE Algorithm and Codebase

We provide the full training algorithm for PolySAE in Algorithm 1, detailing the encoding, polynomial decoding, and optimization steps used throughout all experiments. Our codebase is publicly available at `https://github.com/pakoromilas/PolySAE`

# B. Implementation Details

**Architecture and Sparsification.** We train standard sparse autoencoders (SAEs) and PolySAEs with a latent width of 16,384 and sparsity level $K = 64$. Encoders use one of three sparsification strategies: Top-$K$ (Gao et al., 2025), BatchTopK (Bussmann et al., 2024), or Matryoshka (Bussmann et al., 2025). For Top-$K$ and BatchTopK, the $K$ largest activations per token (or batch) are retained and the remainder zeroed. All models are trained on residual-stream activations extracted from pretrained language models.

**LLMs.** We evaluate SAEs and PolySAEs on a standard set of pretrained language models spanning a range of scales: GPT-2 Small (Radford et al., 2019), Pythia-410M and Pythia-1.4B (Biderman et al., 2023), and Gemma-2-2B (Gemma Team, 2024). For each model, we extract residual-stream activations from a single transformer layer chosen near the center of the network, following the methodology of Dunefsky et al. (2024).

**PolySAE Decoder Ranks.** The rank configurations used in our experiments are:

- GPT-2 Small (Radford et al., 2019): $(768, 64, 64)$

- Pythia-410M (Biderman et al., 2023): $(1024, 128, 128)$

- Pythia-1.4B (Biderman et al., 2023): $(2048, 128, 128)$

- Gemma-2-2B (Gemma Team, 2024): $(2304, 128, 128)$

**Training Setup.** All models are trained using the Adam optimizer with $\beta_1 = 0.9$ and $\beta_2 = 0.999$, a constant learning rate of $3 \times 10^{-4}$ with no warmup or decay schedules. We use a batch size of $4096$ tokens and a context length of $128$. We apply gradient clipping with a maximum norm of $1.0$ to stabilize training. No weight decay or L1 regularization is applied to the encoder or decoder weights. Training runs for $5 \times 10^8$ tokens for Gemma-2-2B and Pythia models, and $3 \times 10^8$ tokens for GPT-2 Small, following the protocol used in the main experiments.

**Datasets.** For Gemma-2-2B and GPT-2 Small, training data is drawn from OpenWebText (Gokaslan et al., 2019). For Pythia-410M and Pythia-1.4B, we use an uncopyrighted variant of the deduplicated Pile (Gao et al., 2021). Reconstruction is evaluated on held-out data from the same distribution as training.

**Evaluation.** We evaluate learned representations using SAEBench (Karvonen et al., 2025). Reported metrics include reconstruction error on held-out data and sparse probing performance on six classification tasks: Bias in Bios (De-Arteaga et al., 2019), AG News (Zhang et al., 2015), EuroParl (Koehn, 2005), GitHub programming languages (CodeParrot, 2022), Amazon Sentiment, and Amazon-15 (Hou et al., 2024).

**Implementation.** Our training pipeline extends `SAELens` (Bloom et al., 2024) to support PolySAE while preserving the standard SAE training interface. PolySAE differs from standard SAEs only in the decoder; all other components, including the encoder, sparsification strategy, optimizer, and evaluation pipeline, are shared across models.

# C. Extended Qualitative Analysis

We present an extended qualitative analysis of the interaction structure learned by PolySAE. The analysis proceeds hierarchically, first examining second-order (pairwise) interactions and then extending to third-order (triplet) compositions. Throughout, we compare PolySAE to a vanilla Top-$K$ SAE trained under identical conditions.

## C.1. Second-Order Analysis

We begin by analyzing pairwise interactions to assess whether PolySAE captures compositional structure beyond surface-level co-occurrence.

**Setup.** Both models are applied to 1M documents from OpenWebText. Features are ranked by total activation mass, and the top 10,000 are retained, yielding approximately $5 \times 10^7$ candidate feature pairs.

**Interaction Strength.** For PolySAE, we quantify the strength of a feature pair $(i, j)$ using the learned quadratic decoder weights:

$$B_{ij} = \lambda_2 \left\| (\boldsymbol{u}_i \odot \boldsymbol{u}_j)^\top \boldsymbol{C}^{(2)} \right\|_2, \tag{5}$$

which reflects how much decoder capacity is assigned to that interaction. For the vanilla SAE, which lacks explicit interaction parameters, we use empirical feature covariance,

$$\text{Cov}_{ij} = \mathbb{E}[z_i z_j] - \mathbb{E}[z_i]\mathbb{E}[z_j], \tag{6}$$

as a proxy for pairwise structure.

**Relation to Co-occurrence.** We independently estimate empirical co-occurrence by counting positions where both features appear in the Top-$K$ active set. For the vanilla SAE, covariance is strongly correlated with co-occurrence ($r = 0.82$), indicating that pairwise structure largely mirrors frequency. In contrast, PolySAE's interaction strengths show negligible correlation with co-occurrence ($r = 0.06$), suggesting that learned interactions reflect structure beyond surface statistics.

**Qualitative Regimes.** The weak coupling between interaction strength and frequency allows us to identify qualitatively distinct regimes. Of particular interest are *latent* interactions: feature pairs with strong learned interactions despite low empirical co-occurrence. These pairs often correspond to meaningful compositional patterns that are not recoverable from frequency alone.

*Table 7.* **Compositional Interactions Captured by PolySAE.** PolySAE features (A and B) bind in context to represent specific compositional concepts. Vanilla SAE features (with high-frequency features filtered) fail to capture these compositions.

| Poly Feature A | Poly Feature B | Context | Vanilla SAE |
|---|---|---|---|
| [star, stars] | [coffee, tea] | We've all certainly heard of beers brewed with espresso, but how about one with an espresso shot poured over the top? **Starbucks** | [Apple, Google] |
| [officially, ically] | [traditional, conventional] | It's hard to say what's most impressive about Eduardo Garcia. The class**ically**-trained chef spent years cooking aboard yachts | [newly, ly] |
| [surgery, repair] | [Trans, LGBT] | Some in the transgender community are worried a suspicious fire at a Montreal clinic will add delays to an already lengthy process to get gender reassignment **surgery** | [birth, baby] |
| [DNA, genetic] | [mod, mods] | Activists are opening up a new front in their campaign against genetic modification. The latest target is genetically-**mod**ified trees, which scientists believe could bring huge sustainability | [modified, edit] |
| [secret, hidden] | [Snowden, WikiLeaks] | On May 24th PBS aired a Frontline documentary about alleged Wikileaker Bradley Manning called "**WikiSecrets**" | [secret, secrets] |
| [business, businesses] | [man, woman] | By Joseph George. The business**man** dad of the boy who drove a Ferrari and was arrested by police in Kerala, India | [man, President] |

**Examples.** For interactions above the 80th percentile in $B_{ij}$, we extract representative contexts in which both features co-activate. We mark the target token in each sentence and label features by their top-activating tokens. Comparing these contexts with vanilla SAE activations highlights cases where PolySAE captures relationships that the linear model does not.

## C.2. Third-Order Analysis

We next examine whether third-order interactions refine or disambiguate pairwise compositions.

**Candidate Selection.** We focus on latent second-order pairs—those with high interaction strength but low co-occurrence—and identify corpus positions where both features are simultaneously active.

**Triplet Scoring.** Within these contexts, we evaluate all co-active third features using the learned cubic decoder:

$$\mathrm{Gamma}(f_1, f_2, k) = \lambda_3 \left| (\boldsymbol{u}_{f_1} \odot \boldsymbol{u}_{f_2}) \cdot \boldsymbol{u}_k^\top \boldsymbol{C}^{(3)\top} \right|. \tag{7}$$

For each pair, we retain the third feature with the highest score, after filtering stopword-like features.

**Interpretation.** The resulting triplets are consistently interpretable, with the third feature modulating the meaning of the pair rather than introducing unrelated content. Common patterns include entity–attribute–domain and subject–object–context structures. Representative examples are shown in Table 12, illustrating how higher-order interactions sharpen and contextualize pairwise compositions.

## C.3. Additional Second-Order Interaction Examples

Tables 7–11 show additional second-order interaction examples. Each row highlights a token where two PolySAE features are simultaneously active. Across these tables, the interacting features are typically more specific than the corresponding vanilla SAE features in the same context. The vanilla SAE often activates on a single high-level, morphological, or broadly related feature, while PolySAE activations reflect a more refined decomposition at the highlighted token.

## C.4. Additional Third-Order Interaction Examples

Tables 12 and 13 present further third-order examples. Each row shows contexts in which three PolySAE features co-activate at the same token. In these cases, the activated features vary with context and appear more specific than the corresponding vanilla SAE activations, which often capture only one component or default to generic features.

*Table 8.* **PolySAE Interactions – Brand & Proper Noun Decomposition.** PolySAE decomposes compound names into their semantic constituents. Vanilla SAE (with high-frequency features filtered) often fires on unrelated entities or only captures the surface form.

| Poly Feature A | Poly Feature B | Context | Vanilla SAE |
|---|---|---|---|
| [economic, economy] | [Times, magazine] | The RBI on Wednesday did not allow Stanley Pignal, the South Asian business and finance correspondent for the **Economist** magazine, to attend the central bank's | [economics, economist] |
| [field, fields] | [University, school] | John Doe is a Jesuit with ADHD. He was an outstanding student and a compassionate senior at Fair**field** University who played sports and volunteered often at a literacy | [York, Washington] |
| [Star, Chronicle] | [staff, crew] | Man Arrested after Stolen Mower Runs Out of Gas. By West Kentucky Star **Staff**. PADUCAH, KY | [staff, faculty] |
| [Dragon, Iron] | [steel, Pittsburgh] | The **Iron** Horde is on the march, and the Warlords of Draenor are primed to invade Azeroth on November 13! Steel yourself for the onslaught by watching | [assault, steel] |
| [Italian, Italy] | [gang, mob] | Details obtained by the Guardian reveal extent to which **Sicilian** mafia clans are migrating north after running into financial problems in Italy. | [State, ISIS] |

*Table 9.* **PolySAE Interactions – Morphological Composition.** PolySAE binds suffix/prefix features with semantic content to form derived words. Vanilla SAE (with high-frequency features filtered) captures only generic morphological patterns without semantic binding.

| Poly Feature A | Poly Feature B | Context | Vanilla SAE |
|---|---|---|---|
| [ers, Workers] | [administration, administrative] | Piedmont High School. A school reveals it has a "Fantasy Slut League" **Administrators** try to do the right thing, but fall woefully short of | [members, ers] |
| [ing, ings] | [arrested, arrest] | Earlier this year, The Heritage Foundation's Meese Center released Arrest**ing** Your Property, a comprehensive report on civil asset forfeiture-the much mal | [ing, training] |
| [protest, protests] | [making, ing] | Major League Baseball can no longer claim to be free of any anthem-protest**ing** players. On Saturday night, A's catcher Bruce Maxwell took | [ing, training] |
| [ers, Workers] | [photos, pictures] | In-Sight Film. The film in-sight was produced in conjunction with the Format Photography Festival to mark 10 years of the Street Photograph**ers** group | [members, ers] |
| [ized, ization] | [treatment, drugs] | The flu shot is a quack science medical hoax. While some vaccines do confer immuniz**ation** effectiveness, the flu shot isn't one of them | [development, ation] |
| [bound, ice] | [gun, guns] | A new Texas law gives gun owners a new right to store a weapon (any lawfully owned firearm, not just those owned under a Concealed Handgun **Lice**nse | [gun, weapons] |

## D. LLM-as-Judge Prompt for Interaction Interpretability

To quantify the interpretability of PolySAE's learned interactions at scale (Section 5.2, Q7), we query GPT-4o-mini with a structured prompt that returns a 0–1 score for how strongly each interaction pair matches the composite concept implied by its two constituent features. For each feature pair $(i, j)$ with non-negligible interaction strength $B_{ij}$, we supply the top-activating tokens of features $i$ and $j$ together with representative corpus contexts in which both features co-activate. The model rates each context independently and returns scores in JSON format. Of the 292,361 pairs with non-negligible $B_{ij}$, we evaluated 70K, of which 8,550 (12%) scored above 0.9. The full prompt is given in Figure 4.

*Table 10.* **PolySAE Interactions – Domain-Specific Collocations.** PolySAE captures specialized terminology through the interaction of domain features. Vanilla SAE (with high-frequency features filtered) often misses the domain-specific meaning.

| Poly Feature A | Poly Feature B | Context | Vanilla SAE |
|---|---|---|---|
| [football, NFL] | [conference, conferences] | Statement from the Southeastern Conference Office Regarding the Florida-LSU football game: The LSU-Florida **football** game scheduled for Saturday in Gainesville | [League, Conference] |
| [earnings, financial] | [number, numbers] | T-Mobile US, Inc. TMUS is scheduled to report fourth-quarter 2015 financial **number**s, before the opening bell on Feb 17. Last | [numbers, figures] |
| [technology, tech] | [development, developers] | You can't look at internet news lately without seeing the latest and greatest in nanotechnology **development**s. Everything these days is being manufactured smaller, faster | [it, said] |
| [Canada, Canadian] | [oil, pipeline] | Eddy Radillo holds a Texas flag and a sign opposing the Transcanada Keystone **Pipeline** in February 2012 outside the Lamar County Courthouse in Paris | [oil, gas] |
| [diet, fitness] | [train, rail] | Here's what you need to know... Your gains will stagnate if you only weight **train** within the same rep ranges and loading patterns. | [training, train] |

*Table 11.* **PolySAE Interactions – Compound Words & Phrases.** PolySAE captures compound words and multi-word phrases through feature interactions. Vanilla SAE (with high-frequency features filtered) often misses the compositional meaning entirely.

| Poly Feature A | Poly Feature B | Context | Vanilla SAE |
|---|---|---|---|
| [director, founder] | [lead, managing] | A FRIEND OF MINE recently made the following observation about Ezra Koenig, the founder and **lead** singer of Vampire Weekend. "Did you realize, | [led, lead] |
| [written, designed] | [research, researcher] | The following story was written and **researched** by Rone Tempest for The Utah Investigative Journalism Project in partnership with The Salt Lake Tribune. Dustin Porter said | [created, made] |
| [alleged, allegations] | [level, levels] | Back to previous page. Accusations against generals cast dark shadow over Army. By Ernesto Londoño. The accusations **lev**eled against | [place, made] |
| [music, musical] | [official, officer] | ROCHESTER, N.Y. – Members of Rochester's music community continue to pull together to remember and help the family a fellow **musician** who | [man, President] |
| [involved, involvement] | [support, help] | STEAL THIS SHOW's Patreon campaign helps keep us free and independent. If you enjoy the show, get **involve**d. Our patrons get access to | [started, ready] |
| [shooting, shot] | [focus, focused] | Berenice Abbott was an American photographer best known for her black-and-white photography of New York City. She heavily focused her **shooting** | [ing, training] |
| [document, documents] | [content, contents] | Use these links to rapidly review the **document**. TABLE OF CONTENTS. INDEX TO CONSOLIDATED FINANCIAL STATEMENTS | [Introduction, History] |

# E. Activation Steering: Additional Examples

We provide additional qualitative examples of activation steering on GPT-2 Small, extending the experiment described in Section 5.2. Table 14 gives the full set of selected examples across the 27 compositional concepts (the main paper shows the first four rows as Table 6). Table 15 isolates cases where vanilla SAE directions fail to shift the output from the unsteered baseline at all, while PolySAE directions produce a meaningful semantic change, illustrating the gap captured by the aggregate +41.5 mean rank improvement in Section 5.2.

*Table 12.* **Third-Order Compositional Interactions Captured by PolySAE.** Three PolySAE features ($F_i$, $F_j$, $F_k$) bind in context to represent compositional concepts. Vanilla SAE often captures individual components but misses the compositional structure.

| Poly $F_i$ | Poly $F_j$ | Poly $F_k$ | Context | Vanilla SAE |
|---|---|---|---|---|
| [nuclear, Fukushima, reactor] | [test, testing, tested] | [radiation, laser, magnetic] | US tests **nuclear**-capable missile with the range to strike North Korea. The US has test-fired a nuclear-capable intercontinental ballistic missile | [nuclear, reactor, atomic] |
| [black, white, racial] | [Americans, Canadians, Australians] | [people, women, men] | In a push to get more Black **Americans** involved in the world of tech, a slew of organizations have teamed up with South by Southwest | [Americans, Muslims, Jews] |
| [ing, ings, ting] | [stock, trading, market] | [investment, invest, investing] | Philippines stocks higher at close of trade; PSEi Composite up 0.57%. **Invest**ing.com — Philippines stocks were higher after | [ing, training, running] |
| [line, lines, lining] | [supply, supplies, shortage] | [road, route, pipeline] | The same is true of supply **lines** into landlocked Afghanistan. Within months of the 2001 invasion, Mr. Musharraf signed a deal | [the, ,, ., ', of] [the, ,, ', of, a] |
| [proved, proven, prove] | [star, stars, superstar] | [reputation, popularity, fame] | Arguably the biggest surprise would have been if he had turned up, but David Bowie proved some **stars** are big enough not to have make themselves available | [star, stars, superstar] |
| [historic, historical, historically] | [UFC, fight, MMA] | [strong, impressive, solid] | After 1,501 days as UFC light-heavyweight champion, Jon Jones' **historic** title reign came to an end late Tuesday when he was stripped | [the, ,, ., ', of] |
| [treated, treat, treating] | [consumers, consumer, consumption] | [customers, customer, clients] | Jeremy Corbyn today warned the banking industry it must not treat **consumers** and entrepreneurs as a "cash cow" and attacked the links between senior politicians | [the, ,, ., ', of] |

```
Give each example sentence a 0 to 1 score for how strongly it matches the composite
concept implied by features i and j.

Feature i representative tokens:  {token_examples_i}
Feature j representative tokens:  {token_examples_j}
Example sentences:  {example_sentences}

Instructions:
- Judge how strongly the intended criterion in the question above is satisfied for
each example sentence.
- 0 means not present at all.
- 1 means very strongly present.
- If there are multiple example sentences, score each one separately.
- Return valid JSON only.
- Do not return any explanation or text outside the JSON.

Return format:
{{
  "scores": [0.0]
}}
```

*Figure 4.* Full LLM-as-judge prompt used in Section 5.2 (Q7) to score the interpretability of PolySAE feature interactions.

## F. Interaction Rank Ablation

Figure 5 examines the effect of interaction ranks on reconstruction, for fixed $R_1 = 768$ on GPT-2 Small. PolySAE achieves competitive reconstruction with modest interaction ranks ($R_2 = R_3 = 64$). Increasing ranks beyond this does not improve reconstruction, suggesting that the additional capacity is unnecessary for capturing interaction structure in this setting.

## G. Computational Overhead

We measure the practical cost of PolySAE's polynomial decoder against a vanilla SAE under identical training conditions on GPT-2 Small ($3 \times 10^8$ training tokens, context length 128, batch size 4096 tokens per step, single GPU). Figure 6 shows reconstruction MSE over training, Figure 7 reports per-step system resource usage, and Table 16 summarizes median resource use and total wall-clock time. PolySAE adds $+1.19$ GB of GPU memory and $+18.2$ min of wall-clock time, with

*Table 13.* **Additional Third-Order PolySAE Interactions.** Further examples of three-way feature compositions. Vanilla SAE sometimes captures individual components but misses the compositional structure.

| Poly $F_i$ | Poly $F_j$ | Poly $F_k$ | Context | Vanilla SAE |
|---|---|---|---|---|
| [Army, Force, Navy] | [Israel, Israeli, Jewish] | [IDF] | Earlier this week, the Friends of the Israel Defense Forces, an organization dedicated to supporting the men and women serving in the **IDF**, held its annual dinner | [Israel, Israeli, Jewish] [the, ,, ., ', of] |
| [annual, monthly, annually] | [percent, %, points] | [regular, regularly, frequent] | According to the latest research from our Wireless Smartphone Strategies (WSS) service, global smartphone shipments grew 6 percent **annually** to reach 360 million units | [annual, monthly, annually] [the, ,, ., ', of] |
| [get, make, getting] | [film, movie, films] | [documented, depicted, depicts] | Three tips on how to **film** anywhere; slums, red light districts, museums, exhibitions, churches, and not get your video camera gear stolen | [film, movie, films] [the, ,, ., ', of] |
| [well, ill, poorly] | [widely, commonly, widespread] | [best, better, good] | April 6, 2014. CR Sunday Interview: Zack Soto. ***** is a widely **well**-liked cartoonist, publisher and | [well, poorly, badly] [the, ,, ., ', of] |
| [accept, accepted, accepting] | [final, ultimate, preliminary] | [great, considerable, significant] | NEW YORK — Dedicated Hillary Clinton supporters accepted **final** defeat Wednesday morning even as they struggled to accept that their candidate lost | [final, finals, ultimate] [the, ,, ., ', of] |
| [percent, %, percentage] | [currency, dollar, euro] | [cents] | The Canadian dollar dipped below 75 **cents** (U.S.) in Tuesday's trading as equity markets worldwide remained extremely volatile | [the, ,, ., ', of] [., $, ,, on, to] |
| [identified, identify, diagnosed] | [virus, Ebola, HIV] | [label, labels, labeled] | the governor of New York State announced that the first case of Ebola had been **diagnosed** at Bellevue | [the, ,, ., ', of] |
| [base, bases, baseline] | [fans, fan, supporters] | [demand, turnout, attendance] | It's a shared problem among fan **bases** across the National Hockey League: They watch their own players so closely that, after a while | [the, ,, ., ', of] [the, ,, ', of, a] |
| [unique, distinct, distinctive] | [two, different, three] | [separate, separated, distinction] | For their collaborative project Jus Now, U.K. producer Sam Interface and Trinidad producer LAZAbeam find singularity in mashing up two **distinct** | [people, men, officers] [the, ,, ., ', of] |
| [line, lines, lining] | [supply, supplies, shortage] | [road, route, pipeline] | The same is true of supply **lines** into landlocked Afghanistan. Within months of the 2001 invasion, Mr. Musharraf signed a deal | [the, ,, ., ', of] [the, ,, ', of, a] |
| [largest, most, biggest] | [able, ible, ability] | [stable, stability, flexible] | Groundwater, the globe's most depend**able** water insurance system, is not as renewable as researchers once thought | [the, ,, ., ', of] [the, ,, ', of, a] |

comparable system RAM and CPU utilization. These overheads are modest relative to the gains reported in Table 1.

*Table 14.* Selected examples on activation steering on GPT-2 Small. We add the first-order decoder directions $\mathbf{d}_i + \mathbf{d}_j$ to the residual stream (layer 8) during greedy generation (27 concepts $\times$ 12 prompts = 324 evaluations per condition). Each row shows the continuation under three conditions: No Steering, SAE, and PolySAE. **Bold** marks where PolySAE steers generation toward the compositional target.

| Features → Target | Prompt | No Steering | SAE | PolySAE |
|---|---|---|---|---|
| [surgery]×[trans] → *gender* | "The procedure that helps individuals align their body with their identity is" | called "*body alignment*." The procedure involves the body aligning... | called "*body alignment*." The procedure involves the use of a combination... | called "**gender identity surgery**." Performed by a surgeon who specializes in gender... |
| [canada]×[oil] → *Keystone* | "The controversial cross-border pipeline project is called the" | *Trans-Pacific Partnership* (TPP), a major step forward for the U.S. and Canada. | *Trans Mountain pipeline*, a controversial project in the works for years. | **Keystone XL**. The pipeline would carry crude oil from Alberta to U.S. refineries. |
| [involved]×[support] → *community* | "The foundation wants more people to become" | *entrepreneurs*, and it wants to make sure they're not just part of the problem. | *entrepreneurs*, and it wants to make sure they're not just part of the problem. | **involved in the community**, and to help them make a difference. |
| [economic]×[times] → *Economist* | "The magazine with coverage of world politics and business is The" | *New York Times*. The magazine with coverage of world politics and business is The New York Times. | *New York Times*. The New York Times is a daily newspaper in the United States... | **Economist**. The Economist is a magazine that is a global news magazine... |
| [alleged]×[level] → *leveled* | "During the hearing criticism was sharply" | *raised* by the government, which said it was not aware of any evidence... | *raised* by the government of India, accused of being "bigoted"... | **leveled** at the government for not doing enough to tackle the problem... |
| [shooting]×[focus] → *product* | "The model arrived early for the scheduled" | launch of the new *iPhone 6 and 6 Plus*, delayed until late March. | launch of the new *iPhone 6 and 6 Plus*, delayed until late March. | launch of the new **Galaxy S8**, delayed until late March. The Galaxy S8 will be the first... |
| [ers]×[admin] → *faculty* | "The university hired new" | *security guards* to guard the campus... | *security guards* to guard the campus... | **faculty members** to help improve its academic performance. |
| [ized]×[treatment] → *vaccination* | "Parents were urged to bring their children in for scheduled" | visits to the hospital after a man was *shot and killed* in a shooting... | visits to the hospital after a man was *shot and killed* in a shooting... | **vaccinations**, but the government has not yet taken action. |
| [secret]×[wikileaks] → *leaks* | "The website that published thousands of diplomatic cables is" | now under fire for publishing a list of the *names of people who leaked them*. | now under fire for publishing a list of the *names of people who leaked them*. | now under fire for publishing a series of **leaked documents** that show the US government...**spy on its own citizens**. |

*Table 15.* Additional activation steering examples where vanilla SAE directions fail to shift the output from the unsteered baseline, while PolySAE directions produce a meaningful change. The first output column shows the shared continuation under no steering and vanilla SAE (both identical); the last column shows the PolySAE-steered continuation. **Bold** highlights the semantic shift introduced by PolySAE.

| Features → Target | Prompt | No Steering = SAE | PolySAE |
|---|---|---|---|
| [document]×[content] → *chapters* | "Before reading the report, check the" | following links: http://www.npr.org/. . . | following: The report states that the ***Department of Homeland Security*** has been working with. . . |
| [Dragon]×[steel] → *Iron* | "The expansion pack introduced a new enemy faction called the" | "*Mage*" that was able to take over the world of the Elder Scrolls Online. | "***Grim Reaper***" that was able to take over the entire map. The Grim Reaper was a powerful enemy. . . |
| [earnings]×[number] → *quarterly* | "The stock moved sharply after the company released its" | *latest earnings report* on Tuesday. The company said it had *lost* $1.4 billion. . . | ***first quarterly results*** since the financial crisis. The company said it had ***sold*** more than $1 billion. . . |
| [Star]×[staff] → *editorial* | "Credit for the investigative piece went to the paper's" | editor, who wrote: "The story was published in the NYT on Monday, and it was a great success." | editor, who wrote: "The paper's ***editorial staff*** has been working hard to ensure that the story is accurate." |
| [shooting]×[focus] → *magazine* | "Behind the scenes the crew was busy with the cover" | of the new *book*, which is due out in September. The book is a collection of stories. . . | of the new ***issue of the magazine***. |
| [involved]×[support] → *political* | "The organization's mission is to get more citizens" | involved in the fight against *climate change*. | involved in the ***political process***. |
| [Dragon]×[steel] → *Dark Age* | "The fantasy novel described warriors who must" | fight against the forces of darkness and the forces of the dark. *[repetitive]* | fight for their lives against the forces of darkness. The novel is set in the world of the ***Dark Age***. . . |
| [ers]×[admin] → *regulatory* | "The organization needed experienced" | and experienced staff to help it grow. *[repetitive]* | staff to help it navigate the ***complex legal and regulatory environment***. |

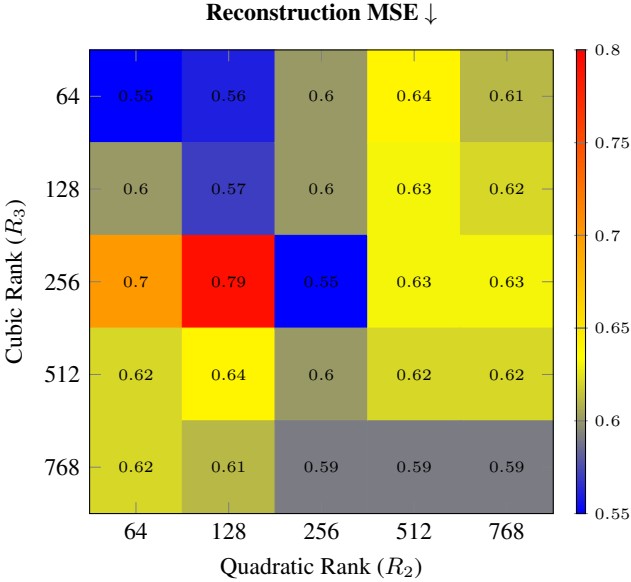

*Figure 5.* Reconstruction MSE for different $R_2$ and $R_3$ values, with $R_1 = 768$, using activations from GPT-2 Small.

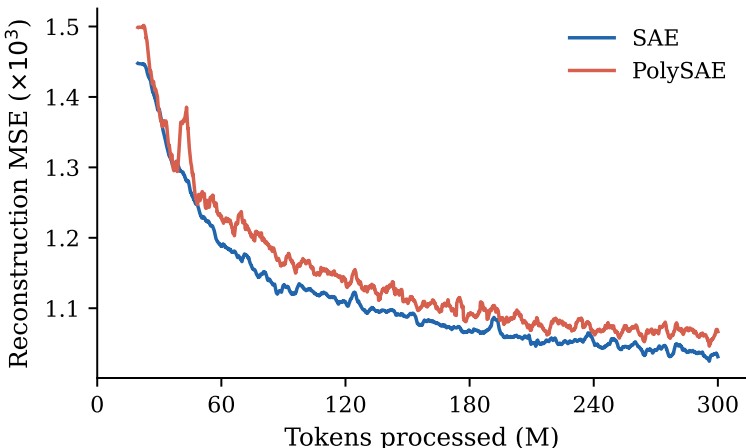

*Figure 6.* Reconstruction MSE over $3 \times 10^8$ training tokens.

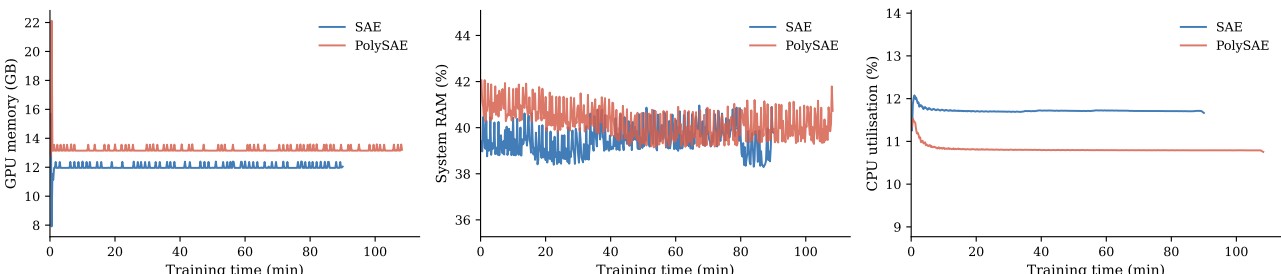

*Figure 7.* GPU memory, system RAM, and CPU utilization recorded during training.

*Table 16.* Median resource usage and wall-clock time during training. $\Delta$ shows absolute difference with relative change in parentheses.

|         | GPU mem. (GB)    | RAM (%)        | CPU (%)        | Wall clock (min) |
|---------|------------------|----------------|----------------|------------------|
| SAE     | 11.96            | 39.4           | 11.7           | 90.0             |
| PolySAE | 13.15            | 40.2           | 10.8           | 108.2            |
| $\Delta$ | +1.19 (+10.0%)  | +0.8 (+2.0%)   | −0.9 (−7.7%)   | +18.2 (+20.2%)   |

