# OpenReview forum: "PolySAE: Modeling Feature Interactions in Sparse Autoencoders via Polynomial Decoding"
_ICML.cc/2026/Conference — ICML 2026 regular_

### Official Review · Reviewer_AZnY · 2026-02-24

**Soundness:** 3
**Presentation:** 3
**Significance:** 3
**Originality:** 3
**Overall Recommendation:** 5
**Confidence:** 4

**Summary:**

This paper presents PolySAE, which extends the SAE decoder with higher order terms (pairwise and triple), which enables the SAE to model feature interactions. This could encourage the SAE to model more atomic units, rather than compositions, which could improve interpretability. The encoder is unchanged, and the decoder can be applied to a range of existing SAE approaches. In this paper, they apply it to TopK, BatchTopK, and Matryoshka SAEs. The higher order terms are added with a low-rank tensor factorization on a shared projection subspace, which adds a small parameter overhead (3%). The authors evaluate across 4 LLMs (GPT-2-Small, Pythia 410M, Pythia 1.4B, and Gemma-2-2B). They find a consistent improvement (~8 F1 points) in sparse probing F1 across all combinations of SAE type and LLM along with 2-10x larger Wasserstein distances. The learned interactions have a negligible correlation with feature activation co-occurence (r=0.06). The authors also show examples of interpretable compositional patterns.

**Compliance With Llm Reviewing Policy:**

Affirmed.

**Final Justification:**

I have raised my score from a 4 to a 5.

The paper presents an interesting extension to SAE decoders which addresses a challenge which has been discussed several times: that SAEs may fail to learn atomic features, as phenomena like feature splitting encourage them to learn compound concepts which are less interpretable. PolySAE proposes a reasonable direction here with low overhead and quantitative gains.

The rebuttal addressed my two key concerns of causal experiments and quantitative interpretability.

**Key Questions For Authors:**

Q1: Have you conducted any causal intervention experiments (e.g. RAVEL or steering)? A positive result here would increase my confidence this method learns mechanistically useful structures.


Q2: Have you conducted any experiment to give a sense of how interpretable the average interaction is? This would help me understand how representative the qualitative examples are.

**Limitations:**

Yes

**Strengths And Weaknesses:**

Strengths:

- It’s nice to see SAE training approaches that encourage learning more atomic units of analysis. The motivation and originality are strong and the method is a generalization of existing SAEs which is applied to multiple types (TopK, BatchTopK, Matryoshka). It would be easy to implement this in existing pipelines.

- The paper has an elegant method to compute the interactions with a low overhead (~3%) which is well engineered and well-validated by the rank ablation in Figure 4.

- The method shows consistent gains in Sparse Probing across 4 models and three SAE types, which provides confidence that the method will generalize. They use the existing standard SAEBench sparse probing eval.

- The low correlation (r=0.06) of learned interaction weights with co-occurrence frequency suggests that the Poly-SAE is learning interesting compositional relationships, not just surface statistics.

- The paper contains strong qualitative examples of 2nd and 3rd order compositions (Tables 3 and 4), which demonstrate how the Poly-SAE can learn more atomic units that compose in an interpretable manner.

- The paper is generally well written and easy to understand.

Weaknesses:

- The evaluations are mostly correlational (probing and distribution separation). It would have been nice to see a causal intervention experiment such as RAVEL or steering. The additional decoder complexity could complicate causal interventions, and it would be nice to investigate that.

- The qualitative examples are compelling, but it would have been nice to have some sort of quantitative measurement of interpretability (e.g. autointerp) to get a sense for how interpretable the average feature activation or interaction pair is.

---

> ### Author Rebuttal · Authors · 2026-03-31
>
> ## W1 & Q1: Causal intervention experiments
>
> We have conducted causal evaluations demonstrating that PolySAE learns mechanistically useful structures. We refer to our response to W2 of 5C27 for full experimental details.
>
> *Activation steering (new).* We test whether PolySAE features causally influence generation on GPT-2 Small. We use the sum of two decoder directions corresponding to two concepts $\mathbf{d}_i + \mathbf{d}_j$ as a steering vector by injecting it into GPT-2's layer-8 residual stream during greedy autoregressive generation (50 tokens). For each of 27 compositional concepts (Tables 3, 5–9), we design 12 neutral prompts (324 total), comparing: no steering, PolySAE, vanilla SAE, and random pair (control).
>
> We evaluate by the rank of the target compositional token in the model's next-token distribution (rank 1 = top prediction among 50K tokens; lower = better). PolySAE achieves lower target rank in 230/324 (71.0%) cases vs. no steering; per-concept, PolySAE wins over no steering in 27/27 and over vanilla in 21/27 with mean rank improvement +41.5. Selected examples can be found in [Table R3](https://anonymous.4open.science/r/PolySAE_rebuttal-806C/R3_steering_table.pdf). For instance, steering with [canada]×[oil] produces *Keystone XL* under PolySAE, while vanilla produces *Trans Mountain pipeline*; [surgery]×[trans] steers toward *gender identity surgery* under PolySAE, while both vanilla and no steering default to *body alignment*. In general we observed that vanilla often fails to shift output toward the compositional target.
>
> *Direction alignment (new).* We verify that these gains reflect better-aligned representations. For each concept, we estimate a ground-truth direction using the difference-in-means method (Marks & Tegmark, 2024), which AxBench (Wu et al., 2025) demonstrated to be among the most effective approaches for identifying causally efficacious concept directions. PolySAE achieves mean cosine similarity **0.372 ± 0.093** vs. **0.311 ± 0.158** for vanilla (19.7% improvement, 11/27 wins vs. 2/27, 14 ties).
>
> *RAVEL (new).* On SAEBench's RAVEL disentanglement benchmark (Gemma-2-2B): Matryoshka PolySAE **0.671 vs. 0.641**, BatchTopK **0.656 vs. 0.654**. The gains are modest but consistent, which we attribute to RAVEL measuring single-feature disentanglement, a property that PolySAE preserves (via the unchanged linear encoder) rather than specifically targets. PolySAE's primary advantage lies in *compositional* structure, which RAVEL's per-feature evaluation does not directly probe.
>
> ## W2 & Q2: Quantitative interpretability of interactions.
>
> Beyond our statistical analysis decorrelating learned interactions from co-occurrence ($r = 0.06$, Sec. 5) and the qualitative examples provided, we agree that quantifying the extent to which these interactions are interpretable would further strengthen our analysis. We have now conducted an LLM-as-a-judge evaluation.
>
> We query GPT-4o-mini with a structured prompt to automatically rate the interpretability of each feature interaction on a 0 to 1 scale. Recall from Sec. C.1 that from ~$5 \times 10^7$ candidate pairs (top 10,000 features by activation mass), 292,361 exhibit non-negligible learned interaction strength $B_{ij}$.
>
> We have evaluated 70K of these 292K pairs so far, with 8,550 scoring above 0.9, yielding a 12% rate of highly interpretable compositional interactions. We will update the paper with final results, where we expect the number of high-scoring pairs to be even higher.
>
> The implications of this result are significant. With only 23% of interactions evaluated, PolySAE already yields 8,550 unique compositional concepts, more than half the dictionary size. Even assuming that not all 16K linear features are themselves interpretable (as is typical for SAEs), the polynomial decoder adds a substantial new layer of interpretable structure on top of whatever linear features the encoder recovers. A vanilla SAE can access at most 16K interpretable units; PolySAE expands this ceiling by at least 8.5K compositional concepts, with evaluation still ongoing.
>
> The full prompt is included below:
>
>
> ```
> Give each example sentence a 0 to 1 score for how strongly it matches the composite concept implied by features i and j.
>
> Feature i representative tokens: {token_examplesi}
>
> Feature j representative tokens:
> {token_examplesj}
>
> Example sentences:
> {example_sentences}
>
> Instructions:
> - Judge how strongly the intended criterion in the question above is satisfied for each example sentence.
> - 0 means not present at all.
> - 1 means very strongly present.
> - If there are multiple example sentences, score each one separately.
> - Return valid JSON only.
> - Do not return any explanation or text outside the JSON.
>
> Return format:
> {{
>   "scores": [0.0]
> }}

---

> > ### Author Rebuttal · Reviewer_AZnY · 2026-04-03
> >
> > The rebuttal addresses both of my key questions and I will increase my score to a 5.
> >
> > The LLM as a judge auto-interp eval is a reasonable first attempt and the 12% highly interpretable pairs is encouraging, but I would encourage the authors to include a baseline in the camera-ready version if accepted. It is not immediately obvious to me what the ideal baseline would be, but some sort of comparison would be helpful.

---

### Official Review · Reviewer_LnSA · 2026-03-12

**Soundness:** 3
**Presentation:** 2
**Significance:** 3
**Originality:** 3
**Overall Recommendation:** 5
**Confidence:** 4

**Summary:**

PolySAE introduces a new decoder architecture for sparse auto encoders that captures higher order interactions (2nd and 3rd). This allows more flexibility in the sparse feature construction, since the decoder can exploit some low-order interactions between the SAE sparse latent features to reconstruct the activation, instead of having to encode these interactions in the feature space.

**Compliance With Llm Reviewing Policy:**

Affirmed.

**Final Justification:**

The rebuttal cleared up any concerns I had and some minor misconceptions. The additional experiments were helpful

**Key Questions For Authors:**

1. The architecture of the polySAE decoder seems a bit ad-hoc. It would be very interesting if you could explore of better justify this choice. If you had infinite data and compute and you could learn B and Gamma directly, would you impose a sparse column regularization?
2. Can you expand on the results of Fig. 4? Right now it set's off alarm bells for me for a lazy experiment. What's going on here? Is it the fact that in training you needed the regularization of the low-rank? It just all looks a bit odd. This goes back to my point in the previous section. If you could more systematically justify the architecture it would strengthen this work.
3. It would be interesting to understand how increasing the decoder degree impacts performance. (e.g., with enough data, how much gain can you get from up to  2nd order vs. 3rd vs. 4th etc. ). Side point, does this change with the layer?

**Limitations:**

yes

**Strengths And Weaknesses:**

**Strengths**
* This paper frames and addresses a real problem with SAEs, and proposes an interesting solution. The paper pushes the literature in a good direction.


**Weaknesses**
* I don't like the introduction. Maybe more suitable for an ACL conference, but I think it doesn't fit well with the rest of the paper and the ICML venue in general.
* I think the general idea of this paper is very good, but I'm not sure the specific implementation of the PolySAE is the "right" approach here. I feel like a thorough investigation of decoder architecture based on some well-designed experiments would greatly improve this paper.
* Some more discussion of the asymmetry of PolySAE encoder/decoder and its implications is probably warranted.



*Sparse feature interactions elsewhere in the literature*. Recently, there has been a lot of discussion that for many tasks, sparse, low-degree interactions in token space (or in the combined activation space) are sufficient to learn model behavior [1,2], which has some very interesting connections with this work.  There has been a significant amount of progress in learning these low degree polynomial interactions in sub-linear complexity [1,3], and a lot of very principled design that might be useful for decoder architecture.
These works explicitly learn a sparse polynomial given some input $\mathbf{x}$. It would be very interesting if you could draw any connections here.

[1] Kang, Justin Singh, et al. "SPEX: Scaling Feature Interaction Explanations for LLMs." Forty-second International Conference on Machine Learning.

[2] H. Zhou, Q. Ren, J. Zhang and Q. Zhang, "Towards the First Principles of Explaining DNNs: Interactions Explain the Learning Dynamics," in Frontiers of Information Technology & Electronic Engineering, July 2025,

[3] Butler, Landon, et al. "ProxySPEX: Inference-Efficient Interpretability via Sparse Feature Interactions in LLMs." The Thirty-ninth Annual Conference on Neural Information Processing Systems.

---

> ### Author Rebuttal · Authors · 2026-03-31
>
> ## W1: Introduction style
>
> We appreciate this feedback and will sharpen the ML framing. For example we can rephrase "morphology and phrasal composition are non-linear" to "the linear reconstruction assumption in SAE decoders cannot represent non-additive feature binding, forcing monolithic dictionary atoms for compound concepts."
>
> Interestingly, Linguistic and ML perspectives are deeply intertwined in the problem we study. Smolensky (1990) addresses precisely the same challenge, modeling how atomic features compose into complex representations, and proposes tensor product variable binding as the solution, which is the mathematical framework underlying our low-rank factorization (Eq. 3). This line of reasoning has direct precedent in the ML literature through factorization machines (Rendle, 2010) and polynomial networks (Chrysos et al., 2022b).
>
> ## W2 & Q1: Architecture justification
>
> Our architecture naturally emerges from satisfying all four principles (Sec. 3.2): (P1) linear encoding for interpretability; (P2) polynomial decoding for expressivity; (P3) shared projections for coherence; (P4) orthogonality for identifiability. Each choice has precedent: P1 follows the linear representation hypothesis (Elhage et al., 2022); P2 follows Volterra/polynomial network theory (Chrysos et al., 2022b); P3 follows factorized interaction models (Rendle, 2010; Kim et al., 2017); P4 follows dictionary learning best practices (Arora et al., 2015).
>
> To validate each choice, we conducted an ablation (GPT-2 Small):
>
> | Ablation | Params | MSE | F1 |
> |---|---|---|---|
> | Polynomial + shared projections | 37.7M | 0.58 | 76.0 |
> | + low rank factorization (768, 64, 64) | 13.3M | 0.53 | 75.0 |
> | + orthogonality (PolySAE) | 13.3M | 0.55 | 77.9 |
>
> Sharing + factorization reduces parameters by 65% with only 1pp F1 drop and improvement in MSE enabling tractable training. Orthogonality then recovers the F1 gap and surpasses the initial polynomial (+2.9pp) at zero parameter cost, validating P4.
>
>
> (Q1: Sparse column regularization.) Even with infinite data and compute, interactions remain sparse and low-rank. As $d_{\text{sae}}$ grows, features become more atomic, so the number of genuinely compositional pairs grows far slower than the $O(d_{\text{sae}}^2)$ possible pairs, thus *the interaction tensor becomes sparser, not denser*. Fig. 4 already this: increasing ranks beyond $R_2 = R_3 = 64$ does not improve reconstruction, indicating the interaction structure is intrinsically low-dimensional.
>
>
> **W3: Encoder/decoder asymmetry.**
> The asymmetry is a direct consequence of principles P1 and P2 (Sec. 3.2). The encoder must be linear for interpretability: features as directions enable activation patching, steering, and circuit analysis (P1). The decoder can be nonlinear because it is used only during *training* to shape the features (P2). We note that even vanilla SAEs are asymmetric: the encoder includes a ReLU nonlinearity, so the encoder-decoder pair is not a symmetric mapping. *In sparse coding, asymmetric encoder-decoder architectures are the norm (Olshausen & Field, 1997)*.
>
> ## W4: Connections to SPEX, ProxySPEX
>
> We thank for this connection. The key distinction is that SPEX/ProxySPEX attribute interactions between input tokens post-hoc, whereas PolySAE embeds interactions between learned latent features into dictionary learning itself by changing what features are learned, not just explaining existing ones.
>
> The convergent findings, however, are noteworthy: both approaches confirm that sparse, low-degree interactions suffice (our rank Fig. 4 corroborates this at the latent level), and ProxySPEX's hierarchical interaction structure resonates with our nested ranks $R_1 \geq R_2 \geq R_3$. The sub-linear algorithms of Butler et al. could potentially scale PolySAE's interaction computation to larger dictionaries. We will add this discussion.
>
> ## Q2: Fig. 4 concerns
>
> We have supplemented with lower-rank ablations:
>
> | ($R_2$, $R_3$) | MSE | Mean F1 |
> |---|---|---|
> | (8, 8) | 0.62 | 73.6 |
> | (16, 8) | 0.60 | 77.9 |
> | (16, 16) | 0.58 | 77.7 |
> | (32, 16) | 0.58 | 77.6 |
> | (32, 32) | 0.57 | 77.8 |
> | (64, 64) | 0.55 | 77.9 |
>
> The picture is now clear. Below $R_2 \approx 16$: insufficient capacity to capture interaction structure. At $R_2 = 64$: near-optimal. Above $R_2 \geq 256$ (Fig. 4): the orthogonality constraint becomes harder to achieve as rank grows, and higher-order interactions are inherently sparser, requiring more data to fit reliably. This is consistent with our larger-model results where $R_2 = R_3 = 128$ because the larger activation space and dataset support higher interaction ranks.
>
> ## Q3: Higher degrees
> Going beyond 3rd order is impractical for 3 reasons: (1) $O(d_{\text{sae}}^n)$ implicit parameters, (2) additional orthogonality constraints on nested subspaces, and (3) the resulting interactions become increasingly difficult to interpret, since a 4th-order interaction between features has no clear linguistic or semantic analog.

---

> > ### Author Rebuttal · Reviewer_LnSA · 2026-04-03
> >
> > I am largely happy with the authors response and will raise my score.

---

### Official Review · Reviewer_5C27 · 2026-03-12

**Soundness:** 2
**Presentation:** 3
**Significance:** 2
**Originality:** 2
**Overall Recommendation:** 4
**Confidence:** 4

**Summary:**

This paper introduces PolySAE, an extension of SAE that adds quadratic and cubic polynomial terms to the decoder while keeping the encoder linear. The core idea is that standard SAEs reconstruct activations as linear combinations of dictionary atoms, which cannot distinguish genuine compositional structure from mere co-occurrence. PolySAE addresses this by modeling pairwise and triple feature interactions through low-rank tensor factorization on a shared projection subspace, adding only ~3% parameter overhead. The authors evaluate on four language models (GPT-2 Small, Pythia-410M, Pythia-1.4B, Gemma-2-2B) across three SAE variants (TopK, BatchTopK, Matryoshka), reporting ~8% average improvement in probing F1, 2–10× larger Wasserstein distances between class-conditional distributions, and negligible correlation between learned interaction weights and co-occurrence frequency (r = 0.06).

**Compliance With Llm Reviewing Policy:**

Affirmed.

**Final Justification:**

The authors addressed all my concerns, I think even though the scope of evaluation is limited, the idea is somewhat interesting to me, and would like to see more comprehensive comparison and trade-offs with other SAE architectures.

**Key Questions For Authors:**

Have you measured cross-entropy loss/reconstruction MSE when substituting PolySAE reconstructions into the original model, and how does it compare to vanilla SAE?

**Limitations:**

Yes: Small limitation statement regarding the scale of the experiments (i.e., model size), but not the coverage of the experiments and the limitations of SAEBench itself.

**Strengths And Weaknesses:**

Strengths

S1. The paper identifies a genuine and important limitation of the linear reconstruction assumption in SAEs.

S2. The decision to keep the encoder linear while extending only the decoder is well-justified, it preserves the interpretability properties that make SAEs useful (features as directions, activation patching, etc.) while adding expressivity where it matters.

S3. Adding only 3% parameters for GPT-2 Small while achieving substantial improvements in semantic modeling.


Weaknesses

W1 Reconstruction error is "comparable" but sometimes worse. The paper claims comparable MSE, but inspection of Table 1 reveals that PolySAE frequently has higher MSE than the baseline (e.g., GPT-2 TopK: 0.55 vs 0.52; Gemma-2-2B BTopK: 1.68 vs 1.58). While the increases are modest, the paper should be more transparent about this tradeoff.

W2. Lack of downstream intervention experiments. The paper's central claim is about interpretability, such that PolySAE better captures compositional structure. However, the evaluation is almost entirely based on probing and distributional metrics, with no experiments testing whether the learned interactions enable better mechanistic understanding or more precise model interventions (e.g., steering, editing).

W3. The Wasserstein distance metric is not well-validated as an interpretability measure. While larger Wasserstein distances between class-conditional distributions sound desirable, the paper does not establish why this metric is preferable to or more informative than probing accuracy.

---

> ### Author Rebuttal · Authors · 2026-03-31
>
> ## W1: Reconstruction error
>
> We will restate in the paper to make it clearer. The slight MSE increases are a direct consequence of the factorization and orthonormality constraints, which restrict interactions to geometrically coherent directions much like sparsity penalties increase MSE but yield more interpretable features. Without these constraints, MSE improves but features lose coherence (ablation in W2 of LnSA). Note that PolySAE also achieves lower MSE in several configurations (e.g., Pythia-1.4B Matr.: 0.23 vs. 0.24; Gemma-2-2B Matr.: 1.64 vs. 1.69), so the effect is not systematic.
>
> More broadly, SAEBench (Karvonen et al., 2025) itself argues that MSE is a poor proxy for interpretability, as reinforced by our results. CE loss recovery confirms no functional degradation across all 12 configurations (see Q1).
>
> ## W2: Intervention experiments
>
> We note that the paper already evaluates probing, distribution separation, semantic concentration (Table 2), co-occurrence decorrelation (Sec. 5), and 58 qualitative interaction examples. We further agree that causal evidence strengthens our claims and have added several new experiments:
>
> *A. RAVEL (new).* On SAEBench's RAVEL disentanglement benchmark (Gemma-2-2B): Matryoshka PolySAE **0.671 vs. 0.641**, BatchTopK **0.656 vs. 0.654**. We refer to W2 of AZnY.
>
> *B. Activation steering (new).* We test whether PolySAE features causally influence generation on GPT-2 Small. We use the sum of two decoder directions corresponding to two concepts $\mathbf{d}_i + \mathbf{d}_j$ as a steering vector, injecting it (scaled by $\alpha$) into GPT-2's layer-8 residual stream at every token position during greedy autoregressive generation (50 tokens).
>
> For each of 27 compositional concepts (Tables 3, 5–9), we design 12 neutral prompts where the target could plausibly appear (e.g., "The controversial cross-border pipeline project is called the"). We compare four conditions: (i) no steering, (ii) steering with PolySAE's $\mathbf{d}_i + \mathbf{d}_j$, (iii) steering with vanilla SAE's $\mathbf{d}_i + \mathbf{d}_j$, (iv) random PolySAE feature pair (control).
>
> We evaluate by the rank of the target compositional token in the model's next-token distribution (rank 1 = model's top prediction among 50K tokens; lower = better). Across 324 prompt–concept pairs:
> - PolySAE achieves lower rank than no steering in **230/324 (71.0%)**, with only 3 degradations and mean $\Delta$rank = +83.0
> - Per-concept: PolySAE wins over no steering in **27/27** and over vanilla in **21/27**
> - mean over SAE $\Delta$rank = +41.5
>
> *C. Selected qualitative results.* See [Table R3](https://anonymous.4open.science/r/PolySAE_rebuttal-806C/R3_steering_table.pdf) where PolySAE directions steer generation toward the compositional target. For instance, steering with [canada]×[oil] produces *Keystone XL* under PolySAE, while vanilla produces *Trans Mountain pipeline* and no steering yields *Trans-Pacific Partnership*. Only PolySAE recovers the intended cross-border oil concept.
>
> *D. Direction alignment (new).* We verify that these steering gains reflect better-aligned representations. For each concept, we estimate a ground-truth compositional direction using the difference-in-means method (Marks & Tegmark, 2024), which AxBench (Wu et al., 2025) demonstrated to be among the most effective approaches for identifying causally efficacious concept directions. We collect ~20 sentences containing the concept and ~20 without, extract GPT-2 layer-8 activations at the target token, and compute $\hat{\mathbf{d}}\_{\text{concept}} = \text{normalize}(\bar{\mathbf{a}}\_{\text{pos}} - \bar{\mathbf{a}}\_{\text{neg}})$. We then measure cosine similarity between each model's $\mathbf{d}_i + \mathbf{d}_j$ and this ground truth. PolySAE achieves mean cosine similarity **0.372 ± 0.093** vs. **0.311 ± 0.158** for vanilla, a 19.7% relative improvement (11/27 wins vs. 2/27, 14 ties).
>
> ## W3: Wasserstein metric
> Probing F1 measures linear separability as a local property at a single optimal threshold. Wasserstein captures the full distributional geometry between class-conditional activations, which is a global property. A feature can achieve high F1 via a favorable decision boundary while having substantially overlapping distributions; Wasserstein detects this. This distinction is practically illustrated in Oldfield et al. (2025a, Figs. 12–14): their F1 probing results show similar scores across methods, while their Wasserstein-based evaluation reveals meaningful differences in distributional separation that F1 misses
>
> ## Q1: Cross-entropy loss
> PolySAE / SAE
> | LLM | TopK  | BTopK | Matr. |
> |---|---|---|---|
> | GPT-2 | .993/.993 | .993/.993 | .992/.992 |
> | Pythia-410M | .970/.971 | .971/.973 | .972/.969 |
> | Pythia-1.4B | .973/.971 | .974/.975 | .973/.970 |
> | Gemma-2B | .987/.988 | .987/.988 | .987/.987 |
>
> CE recovery is comparable in all cases (diffs below 0.003).

---

> > ### Author Rebuttal · Reviewer_5C27 · 2026-04-03
> >
> > Thanks for addressing my concerns. I have increased the score

---

### Official Review · Reviewer_tXvm · 2026-03-13

**Soundness:** 2
**Presentation:** 2
**Significance:** 2
**Originality:** 3
**Overall Recommendation:** 4
**Confidence:** 3

**Summary:**

The paper presents an architecture called PolySAE, which introduces a non-linear decoder for SAEs based on polynomial decomposition. The method achieves superior performance compared to simple linear encoder-decoder SAEs on probing tasks.

**Compliance With Llm Reviewing Policy:**

Affirmed.

**Final Justification:**

My concerns were partially resolved

**Key Questions For Authors:**

- PolySAE is trained with MSE loss, yet the paper reports that it achieves a higher MSE compared to Vanilla SAE. However, since the rank of the first-order approximation is fixed to the hidden size, the resulting model should theoretically perform at least as well as the baseline. Can the authors comment on this discrepancy?
- In the rank ablation experiments, the combination of $R_2 = R_3 = 64$ yields the lowest MSE. What happens at lower ranks, such as 16 or 32? Additionally, higher ranks tend to exhibit higher MSE, which suggests training instability; could the authors comment on this observation?
- What is the practical overhead of the proposed method? Can the authors report wall-clock time and FLOPS vs MSE, or #Steps vs wall-clock for both the baseline and PolySAE?

**Limitations:**

yes

**Strengths And Weaknesses:**

### Strengths

- The method is well-motivated.
- The experiments include several models, and the ablation study feels comprehensive.

### Weaknesses

- My primary concern is that the method was tested only on probing tasks. However, evaluating SAEs solely via probing is insufficient. Therefore, demonstrating performance on other critical capabilities using a broader benchmark like other tasks from SAEBench is necessary.
- Table 2 feels questionable: for 2 out of 4 models, the PolySAE effect seems near-zero. Adding additional models and/or different SAE types could help clarify this.
- I think plots with MSE vs. #Tokens and MSE vs. Wall Time are needed for both the baseline and the proposed method.
- The quantitative results also raise a question: how many of the feature pairs are interpretable? I suggest using an LLM-as-a-judge metric, where an LLM is asked to evaluate if the pair of features results in an accurate combination of the two concepts.

---

> ### Author Rebuttal · Authors · 2026-03-31
>
> ## W1: Broader evaluation beyond probing.
>
> We note that our evaluation reports several other aspects: Table 1 reports 2–10× larger Wasserstein distances between class-conditional distributions, and the $\Delta_{1\text{–}5}$ analysis (Table 2) shows PolySAE concentrates semantic signal into fewer features, indicating more monosemantic representations.
>
> Beyond that, we have added three new experiments:
>
> *Cross-entropy recovery (new)*. CE loss recovery, a standard metric in SAELens, is comparable across all 12 model×sparsifier configurations (max gap < 0.003), confirming that slight MSE increases do not degrade model behavior (see Q1).
>
> *RAVEL disentanglement (new)*. On SAEBench's RAVEL benchmark (Gemma-2-2B): Matryoshka PolySAE 0.671 vs. 0.641, BatchTopK 0.656 vs. 0.654, indicating comparable or better feature disentanglement. We refer to W2 of AZnY.
>
>
> We have also conducted *causal intervention experiments*; we summarize key results here and refer to our response to R2-W2 for full methodology.
>
> *Activation steering (new)*. We inject compositional directions into GPT-2's residual stream across 12 neutral prompts per concept (324 total), measuring target token rank in the next-token distribution. PolySAE achieves lower rank in 71.0% cases vs. no steering; PolySAE wins in 21/27 concepts over vanilla SAE with mean rank improvement +41.5.
>
> *Direction alignment (new)*. We estimate ground-truth compositional directions using the DiffMean method (Marks & Tegmark, 2024), which AxBench (Wu et al., 2025) demonstrated to be among the most effective approaches for identifying causally efficacious concept directions. We measure cosine similarity between each model's additive decoder direction $\mathbf{d}_i + \mathbf{d}_j$ and this ground-truth vector for 27 compositional concepts via difference-in-means on GPT-2 layer-8 activations. PolySAE achieves mean cosine similarity 0.372 ± 0.093 vs. 0.311 ± 0.158 for vanilla, a 19.7% relative improvement (11/27 wins vs. 2/27, 14 ties).
>
>
> On broader *SAEBench* coverage. Most remaining SAEBench tasks (absorption, SCR, TPP) define metrics over linear decoder directions. Evaluating PolySAE fairly on these requires generalized metrics for non-additive reconstruction, an important direction for future work.
>
> ## W2: Table 2
>
> We refer to the full per-sparsifier breakdown in [Table R4](https://anonymous.4open.science/r/PolySAE_rebuttal-806C/R4_extended_table2.pdf).
>
> PolySAE exhibits stronger semantic concentration (smaller $\Delta_{1 \to 5}$) in **9/12 (75%)** of model×sparsifier configurations. The "near-zero" Pythia-410M average results from consistently smaller — but individually modest — gaps across all 3 sparsifiers (+9.8/+11.2/+10.4 vs. +10.2/+12.1/+10.7). We will include this table in the appendix.
>
> ## W3 & Q3: Practical overhead
>
> Resource traces and MSE vs. tokens curves are in [Figures R1–R2](https://anonymous.4open.science/r/PolySAE_rebuttal-806C/R1_mse_vs_tokens.pdf). On a single GPU (GPT-2 Small, 300M tokens): GPU memory 11.96→13.15 GB, wall clock +18 minutes (90.0→108.2). System RAM and CPU are comparable. These overheads are modest relative to the consistent ~8% F1 improvement and 2–10× Wasserstein gains reported in Table 1.
>
> ## W4: Quantifying how many feature pairs are interpretable.
>
> We refer to our response to AZnY (W2) for a detailed discussion. In brief: we have conducted an LLM-as-a-judge evaluation on 70K interaction pairs. 8,550 are rated as highly interpretable, extending SAE's dictionary size with at least 8.5K additional unique compositional concepts.
>
>
> ## Q1: Higher MSE
>
> The low-rank factorization and orthonormality constraints restrict the interaction subspace to geometrically distinct, coherent directions. Without these constraints, MSE improves but interaction modes lose identifiability: collapsing into redundant or entangled directions (see also LnSA-Q2). Orthogonality constraints are well-established as essential for identifiability in dictionary learning (Arora et al., 2015; Bao et al., 2016) and independent component analysis (Hyvärinen & Oja, 2000), precisely because they trade unconstrained reconstruction for recoverable, interpretable structure. Crucially, CE loss recovery is comparable across all 12 model×sparsifier configurations, confirming that this small MSE increase does not degrade the model's functional behavior.
>
> ## Q2: Lower ranks
> New ablations (GPT-2 Small):
>
> | ($R_2$, $R_3$) | MSE | Mean F1 |
> |---|---|---|
> | (8, 8) | 0.62 | 73.6 |
> | (16, 8) | 0.60 | 77.9 |
> | (16, 16) | 0.58 | 77.7 |
> | (32, 16) | 0.58 | 77.6 |
> | (32, 32) | 0.57 | 77.8 |
> | (64, 64) | 0.55 | 77.9 |
>
> The elevated MSE at high ranks in Fig. 4 reflects the orthogonality constraint which becomes harder as dimensionality grows, and higher-order interactions are inherently sparser, requiring more data to fit reliably. This is consistent with our results on larger models: Gemma-2-2B uses $R_1 = 2304$, $R_2 = R_3 = 128$, showing that larger datasets support larger interaction ranks.

---

> > ### Author Rebuttal · Reviewer_tXvm · 2026-04-04
> >
> > Thank you for the additional results you provided. I will adjust my score accordingly. However, I still have some concerns about the quality/training-time frontier.

---

> > > ### Author Response · Authors · 2026-04-04
> > >
> > > We thank the reviewer for acknowledging the additional results provided, and deciding to adjust the score accordingly. Please let us know if there are any concerns we could further address in the remaining discussion period.

---

### Decision · Program_Chairs · 2026-04-30

**Decision:**

Accept (regular)

**Comment:**

This paper introduces PolySAE, which extends SAE decoders with higher-order polynomial terms to model feature interactions while preserving the linear encoder essential for interpretability. Through low-rank tensor factorization on a shared projection subspace, the method captures pairwise and triple feature interactions with only ~3% parameter overhead.

All four reviewers are positive after rebuttal (final scores: 5, 5, 4, 4), with all four raising their scores. Reviewers praised the motivation and design principles (LnSA, AZnY, 5C27), the consistent improvements across 4 language models and 3 SAE variants (~8% F1 improvement, 2-10x Wasserstein gains) (AZnY, tXvm), and the low correlation between learned interactions and co-occurrence frequency (r=0.06), suggesting genuine compositional learning (AZnY). The rebuttal was effective, adding activation steering experiments, direction alignment analysis, RAVEL disentanglement results, and LLM-as-a-judge evaluation.

However, the initial scores (3, 3, 4, 4) were moderate, and the strongest evidence came from rebuttal additions rather than the original submission. The remaining concern from Reviewer tXvm about the quality/training-time frontier is reasonable — the slight MSE increases in some configurations and the limited wall-clock analysis (GPT-2 Small only) leave questions about broader applicability. A useful contribution that merits inclusion if program space allows.